# Optical mapping of ground reaction force dynamics in freely behaving *Drosophila melanogaster* larvae

**Jonathan H Booth**[1,2,3,4], **Andrew T Meek**[1,2,4], **Nils M Kronenberg**[1,2,4], **Stefan R Pulver**[3,4]*, **Malte C Gather**[1,2,4]*

[1]SUPA, School of Physics and Astronomy, University of St Andrews, St Andrews, United Kingdom; [2]Humboldt Centre for Nano- and Biophotonics, Department of Chemistry, University of Cologne, Cologne, Germany; [3]School of Psychology and Neuroscience, University of St Andrews, St Andrews, United Kingdom; [4]Centre of Biophotonics, University of St Andrews, St Andrews, United Kingdom

**\*For correspondence:**
sp96@st-andrews.ac.uk (SRP);
malte.gather@uni-koeln.de
(MCG)

**Competing interest:** The authors declare that no competing interests exist.

**Abstract** During locomotion, soft-bodied terrestrial animals solve complex control problems at substrate interfaces, but our understanding of how they achieve this without rigid components remains incomplete. Here, we develop new all-optical methods based on optical interference in a deformable substrate to measure ground reaction forces (GRFs) with micrometre and nanonewton precision in behaving *Drosophila* larvae. Combining this with a kinematic analysis of substrate-interfacing features, we shed new light onto the biomechanical control of larval locomotion. Crawling in larvae measuring ~1 mm in length involves an intricate pattern of cuticle sequestration and planting, producing GRFs of 1–7 µN. We show that larvae insert and expand denticulated, feet-like structures into substrates as they move, a process not previously observed in soft-bodied animals. These 'protopodia' form dynamic anchors to compensate counteracting forces. Our work provides a framework for future biomechanics research in soft-bodied animals and promises to inspire improved soft-robot design.

## eLife assessment

This study reports **important** findings about new locomotory dynamics of crawling *Drosophila* larva based on imaging the reaction forces during larval crawling. The evidence with the new high-resolution microscopy method is **compelling** as it significantly improves the spatial, temporal, and force resolution compared to previous methods for studying *Drosophila* larva and could be applied to other crawling organisms. The article explains the new technology, WARP microscopy, and provides analysis of the data to characterise small animal behaviour and discover new crawling-associated anatomical features and motor patterns. The work will be of interest to the broad neuroscience community interested in the mechanisms of locomotion in a highly tractable model.

## Introduction

Locomotion is a fundamental behaviour in the Animal Kingdom. There is great diversity in how it is accomplished, from the modification of torque angles in rigid-bodied animals (*Audu et al., 2007*) to a diverse array of peristalses in limbed (*van Griethuijsen and Trimmer, 2014*) and limbless soft-bodied animals (*Berrigan and Pepin, 1995*). Key to these different strategies is one unifying characteristic: action against a substrate or fluid produces forces, thereby translating the body in space. In an aquatic environment, forces acting within fluids can be visualised via the waves of distortion they cause, thus

facilitating the development of detailed theories of movement (*Gray and Llissmann, 1964*). In terrestrial settings, however, substrates are often rigid and therefore prevent direct visualisation of the ground reaction forces (GRFs) generated by animals.

Interactions with substrates have been extensively studied in animals with articulating skeletons (i.e. rigid-bodied animals) due to the ability to calculate output forces from lever physics combined with measurements of joint angles (*Audu et al., 2007*; *Bobbert et al., 2007*). However, much less is known about substrate interactions and GRFs in soft-bodied animals without rigid internal or external skeletons. These animals lack articulating joints upon which muscles act, ambiguating points through which the animal interacts with the substrate. However, they too must anchor a part of their body when another part is in motion to prevent net progression being impeded by an equal but opposite reaction force, that is, their movements must obey Newton's third law of motion (*Trueman, 1975*). Furthermore, soft bodies pose a difficult control problem owing to their highly nonlinear physical properties and virtually unlimited degrees of freedom. Movement over terrain therefore presents a unique challenge for soft animals. Dynamic anchoring has long since been postulated to be at the heart of soft-bodied locomotion (*Tanaka et al., 2012*), but understanding the mechanisms by which soft animals achieve this remains an open problem. Prior work on caterpillars (*van Griethuijsen and Trimmer, 2014*; *Lin and Trimmer, 2010*; *Lin and Trimmer, 2012*; *Lin et al., 2011*), leeches (*Cacciatore et al., 2000*; *Kampowski et al., 2016*) and *Caenorhabditis elegans* (*Fang-Yen et al., 2010*; *Gjorgjieva et al., 2014*) provided key insights and have provided foundational observations for the inspiration of soft robot design; however, a lack of methods with sufficient spatiotemporal resolution for measuring GRFs in freely behaving animals has limited progress.

However, in the field of cellular mechanobiology, many new force measuring techniques have been developed which allow measurement of comparatively small forces from soft structures exhibiting low inertia (*Krieg et al., 2019*; *Mohammed et al., 2019*; *Zancla et al., 2022*) often with relatively high spatial resolution. Early methods such as atomic force microscopy (AFM) required the use of laser-entrained silicon probes to make contact with a cell of interest (*Krieg et al., 2019*). This approach is problematic for studying animal behaviour due to the risk of the laser and probe influencing behaviour. Subsequently, techniques have been developed which allow indirect measurement of substrate interactions. One such approach is traction force microscopy (TFM) in which the displacement of fluorescent markers suspended in a material with known mechanical properties relative to a 0-force reference allows for indirect measurement of horizontally aligned traction forces (*Zancla et al., 2022*; *Lekka et al., 2021*; *Yang et al., 2006*). This technique allows for probe-free measurement of forces, but has insufficient temporal resolution for the measurement of forces produced by many behaving animals, despite recent improvements (*Li et al., 2021*). A second approach revolves around the use of micropillar arrays; in this technique, horizontally aligned traction forces are measured by observing the deflection of pillars made of an elastic material with known mechanical properties. This approach provides excellent temporal resolution but with limited spatial resolution (*Schoen et al., 2010*; *Gupta et al., 2015*).

Recently, we introduced a technique named elastic resonator interference stress microscopy (ERISM) which allows for the optical mapping of vertically aligned GRFs in the nanonewton range with micrometre precision by monitoring changes in local resonances of soft and deformable microcavities. This technique allows reference-free mapping of substrate interactions as well as calculation of vertically directed GRFs used in cell migration (*Kronenberg et al., 2017b*; *Liehm et al., 2018*; *Dalaka et al., 2020*). Until recently, this technique was limited by its low temporal resolution (~10 s), making it unsuitable for use in recording substrate interaction during fast animal movements, but a very recent further development of ERISM known as wavelength alternating resonance pressure (WARP) microscopy has been demonstrated to achieve down to 10 ms temporal resolution (*Meek et al., 2021*). Given ERISM and WARP allow for probe-free measurement of vertical GRFs with high spatial and now temporal resolution, it becomes an attractive method for animal-scale mechanobiology.

In parallel, great strides have been made in understanding the neural and genetic underpinnings of locomotion in the *Drosophila* larva (*Fushiki et al., 2016*; *Zwart et al., 2016*; *Schneider-Mizell et al., 2016*; *Pulver et al., 2015*; *Heckscher et al., 2012*), a genetically tractable soft-bodied model organism (*Brand and Perrimon, 1993*). *Drosophila* larvae are segmentally organised peristaltic crawlers that move by generating waves of muscle contractions (*Berrigan and Pepin, 1995*; *Heckscher et al., 2012*). Larvae have segmentally repeating bands comprised of six rows of actin trichomes (denticles)

(*Payre, 2004*). The developmental and genetic origins of these structures have been extensively studied, but relatively little is known about how they are articulated during movement. While computational modelling and biomechanical measurements have provided an initial knowledgebase (*Loveless et al., 2019*; *Loveless et al., 2021*; *Gjorgjieva et al., 2013*), data on biomechanical forces generated during substrate interactions in *Drosophila* larvae remain extremely limited (*Khare et al., 2015*; *Sun et al., 2022*). Development of methods for measuring GRFs in this model organism would enable fully integrated neurogenetic-biomechanical approaches to understanding soft-bodied movement and fulfil calls from the modelling community for more biomechanics data (*Tytell et al., 2011*).

Here, we develop ERISM- and WARP-based approaches to measure GRFs exerted by freely behaving *Drosophila* larvae. We combine these measurements with kinematic tracking to explore how soft-bodied animals overcome fundamental biophysical challenges of moving over terrain. We find that, despite their legless appearance, *Drosophila* larvae interact with substrates by forming and articulating foot-like cuticular features ('protopodia') and cuticular papillae, which act as dynamic, travelling anchors. The use of ERISM-WARP provides a step change in capability for understanding how soft-bodied animals interact with substrates and paves the way for a wider use of optical force measurement techniques in animal biomechanics and robotics research.

## Results
### Kinematic tracking of substrate-interfacing features

As a first step in understanding how larvae interact with substrates, we confined third-instar larvae to glass pipettes lined with soft agarose (0.1% w/v) (*Figure 1a*). This allowed us to laterally image the animals and the lateral edges of denticle rows at the substrate interface (*Figure 1b*, *Video 1*) while animals crawled towards an appetitive odour source. Animals interact with the substrate by large, soft, segmentally repeating cuticular features that contain rows of denticles and to which we refer as 'protopodia' in the following. Protopodia in each segment engaged in 'swing' periods (moving, SwP) and 'stance' periods (planted on substrate, StP) as waves propagated through the body. During SwPs, protopodia detached from the substrate, with the posterior row of denticles moving to meet the anterior row of denticles, thereby inverting the cuticle and sequestering the whole protopodia into a travelling pocket (*Figure 1c*). When protopodia ended their SwP, they unfolded from the sequestration pocket and then protruded into the substrate during the StP.

To further investigate the dynamics of protopodia placements, we performed detailed kinematic tracking of the morphometry of protopodia, denticle bands, and inter-protopodial spaces during peristaltic waves. By tracking the movement of defined points on bands relative to each other, we monitored intersegmental and intra-segmental movements during peristaltic waves (*Figure 2a*, *Video 2*). In addition to moving relative to each other, denticle bands changed their shape during the sub-phases of a peristaltic wave. During forward waves (peristaltic contractions travelling in an anterograde direction), the anterior-most row of each denticle started to move after the corresponding posterior-most row (*Figure 2b*) and completed its movement after the posterior-most row stopped moving (*Figure 2c*), that is, there was an anteroposterior (AP) latency for both swing initiation (SI) (when movement begins) and for swing termination (ST) (when movement ends). Such a 'rolling' progression pattern is analogous to the 'heel-to-toe' footfalls of limbed animals (*Federle and Labonte, 2019*). To analyse this pattern further, we quantified the percentage of the wave duration spent in AP latency during SI and ST. For forward waves, this relative latency was generally consistent across the denticle bands on large protrusive protopodia but less pronounced for the smaller and less protruding protopodia at the extreme posterior and anterior abdomen and the thorax (*Figure 2d*). In backward waves, the heel-toe-like latency was reversed, with anterior-led latencies observed in SI and posterior-led latencies observed in ST (*Figure 2—figure supplement 1*).

In summary, each segment-wise denticle action event is composed of four distinct periods: SI, SwP, ST, and StP. For forward waves and posterior segments, the latencies during the SI period are largely determined by wave duration ($R^2$ range: 0.46–0.78, A7-A4) but this is less the case for anterior abdomen and thorax ($R^2$ range: 0.12–0.35, A3-A1 and T3, *Figure 2e*). The magnitudes of ST-related latencies are not strongly determined by wave duration ($R^2$ range: 0.01–0.26, *Figure 2f*).

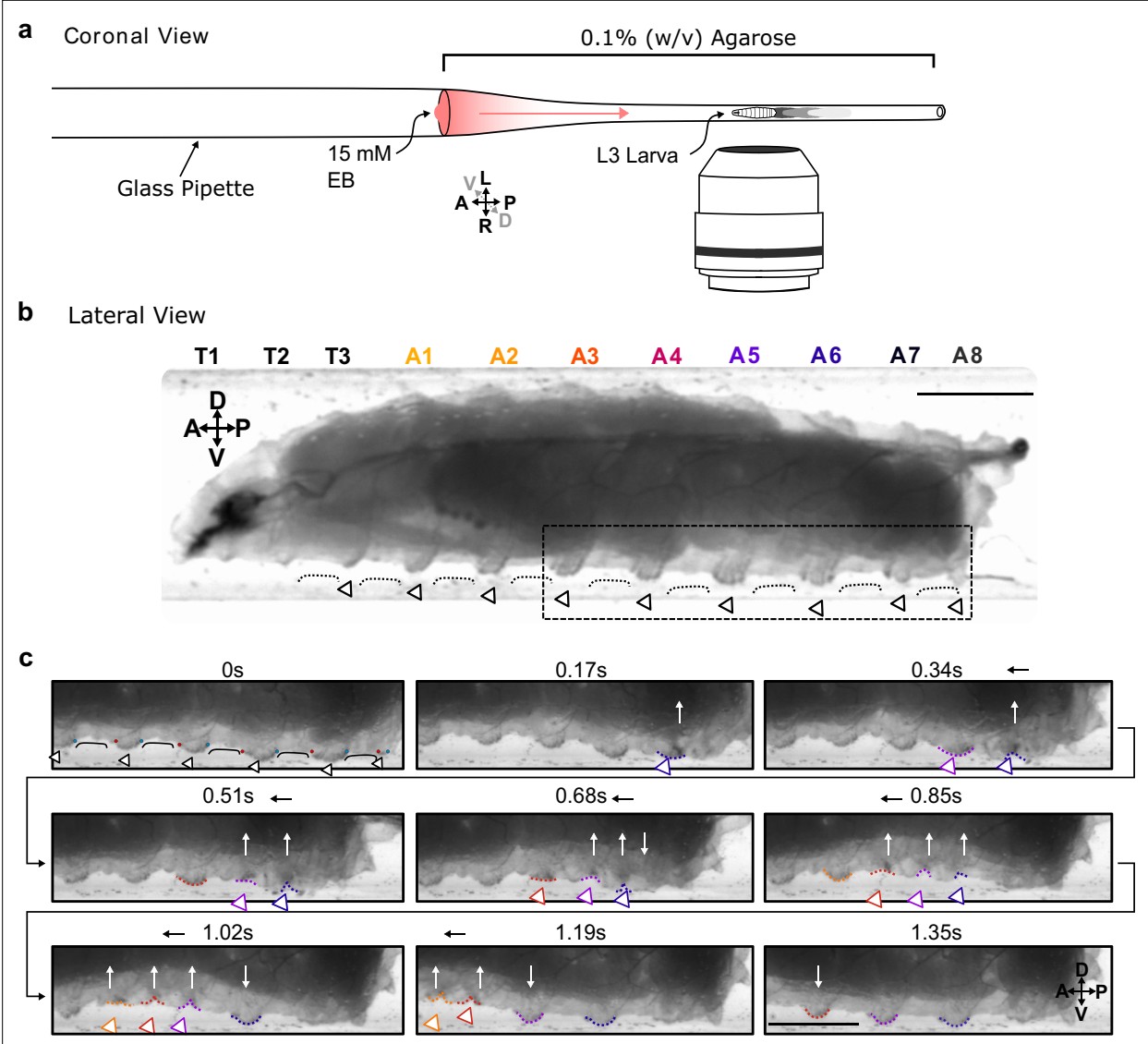

**Figure 1.** Protopodia protrusions in each segment are sequestered during swing phases of forward locomotion. (**a**) Schematic of setup for lateral imaging of larvae, using confinement in Pasteur pipette pre-filled with 0.1% (w/v) agarose. To encourage forward crawling, 10 µL of 15 mM ethyl butanoate (EB) was placed as attractive odour at the end of the pipette. (**b**) Lateral brightfield image of third-instar larva showing convex areas of denticle bands (open arrowheads) protruding into the substrate, interdigitated by concave areas of naked cuticle (black line) not interacting with the substrate. Scale bar = 750 µm. (**c**) Time lapse of area marked by dotted box in (**b**) showing the swing periods and stance periods of protopodia (coloured open arrowheads and dotted lines) during a forward wave. Red and blue dots at 0 s denote anterior and posterior rows of denticles, respectively. As the posterior-most denticle row moved to meet the anterior row of the band, the medial row detached from the substrate via invagination (white arrows). The invaginated pocket is then moved forwards (black arrow) and subsequently replanted. This action repeats as the wave propagates. Scale bar = 500 µm. Images representative of three third-instar larvae.

## Developing stress microscopy for *Drosophila*

Kinematic analysis of protopodia movements revealed a previously uncharacterised complexity in the dynamics of larval movement, but it cannot quantify the mechanical forces impacting the substrate and is therefore limited to making inferences regarding substrate interaction. To achieve quantitative observations, we therefore adapted ERISM-WARP (*Figure 3a*, *Figure 3—figure supplement 1*) to map the vertically directed GRFs exerted by larvae rather than the forces exerted by single cells. First, we developed optical microcavities with mechanical stiffnesses in the range found in hydrogel substrates commonly used for studying *Drosophila* larval behaviour, that is, Young's modulus (E) of 10–30 kPa (*Ahearne et al., 2005*; *Salerno et al., 2010*; *Apostolopoulou et al., 2014*). These

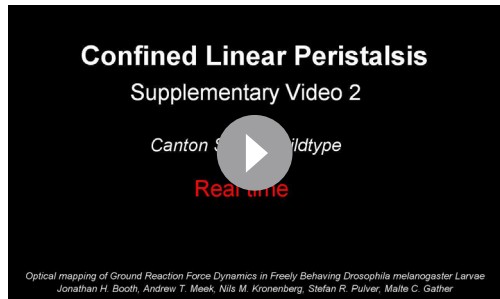

**Video 1.** Lateral view crawling. Video showing the sequestration and planting of protopodia during locomotion from a lateral view.

https://elifesciences.org/articles/87746/figures#video1

microcavities consisted of two semi-transparent, flexible gold mirrors sandwiching a transparent polymer rubber that was made from a mixture of siloxanes with discrete Young's moduli to adjust the resulting stiffness (*Palchesko et al., 2012*). The microcavities were characterised using AFM and the resulting force distance curves (*Figure 3b*) were fitted to a height-corrected Hertz model to determine the Young's modulus of each cavity (*Dimitriadis et al., 2002*). This procedure allowed us to fabricate microcavities with a wide range of well-defined Young's moduli (*Figure 3c*, *Supplementary file 1*).

As an initial test, we placed cold-anaesthetised second-instar larvae onto a microcavity (E = 28 kPa) and performed ERISM force mapping at different magnifications to record substrate indentations generated by larval body features (*Figure 3d–l*). Indentation maps were computed from the images of optical interference by pixelwise solving of the resonance condition with an optical model. Stress maps were then computed from the indentation maps via a finite element method (FEM) simulation of the stress distribution required to produce the observed indentation profile ('Materials and methods'; the accuracy of our calculations was confirmed applying a known force with an AFM, *Figure 3—figure supplement 2*). With this approach, we were able to resolve indentations from rows of denticle bands interdigitated by naked cuticle (*Figure 3g–i*). At higher magnification and when using slightly softer microcavities (E = 19 kPa), even indentations from individual denticles within these bands were resolved (*Figure 3j–l*). The median force exerted by individual denticles was 11.51 nN (1.4–47.5 nN; n = 130 denticles) across a median area of 2.81 μm (1.15–9.13 μm; n = 130 denticles).

## Videorate force mapping in freely behaving animals

Next, we moved to force mapping of freely behaving animals. First, we confirmed that ordinary larval behaviour is maintained on collagen-treated microcavity substrates (*Figure 4—figure supplement 1*). We then adapted WARP (*Meek et al., 2021*) to image substrate interactions at high temporal resolution (*Figure 4—figure supplement 2*). For forward peristaltic waves, we observed posterior to anterior progressions of indentations into the cavity, corresponding to protopodial placements (*Figure 4A*, *Video 3*). We also observed upward deflections of the substrate (i.e. increase in microcavity thickness, positive stress), associated with the displacement of elastomer because of Poisson's ratio governing elastic materials (*Pritchard et al., 2013*). We also observed that the animals travel surrounded by a relatively large water droplet. During StP, protopodia displaced the substrate, and during SwP, protopodia local to the contraction were completely removed from the substrate while travelling to their new resting position.

We also used WARP to investigate the bilaterally asymmetric headsweeps generated by *Drosophila* larvae to sample odours and direct navigation. During headsweeps, anterior segments and mouth hooks detached or dragged across the substrate before replanting (*Figure 4b*, *Video 4*). 0.5–1 s prior to headsweep initiation, the contact area in posterior segments increased, spreading outwards laterally, employing both the protopodia and the naked cuticle along the midline (*Figure 4c*). This broad but shallow anchoring quickly returned to the ordinary resting phase profile after the mouth hooks were replanted onto the substrate (*Figure 4d*).

Before forward waves and headsweeps, larvae produced large indentations posterior to their terminal segment. Anatomical examination revealed accessory structures located at the terminus of the posterior abdomen. Together with the terminal denticle band, these cuticular processes generated tripod-shaped indentation patterns (*Figure 4e*). The left and right sides of the tripod deployed and detached simultaneously (*Figure 4f*). Tripod formation was seen before all observed forward waves (n = 28 across six animals) and bilateral thoracic activity (n = 3 across two animals), but not all tripod contacts resulted in further behaviour (*Figure 4g*). To investigate the relationship between tripod placement and locomotion further, we recorded the delay between tripod contact and protopodial

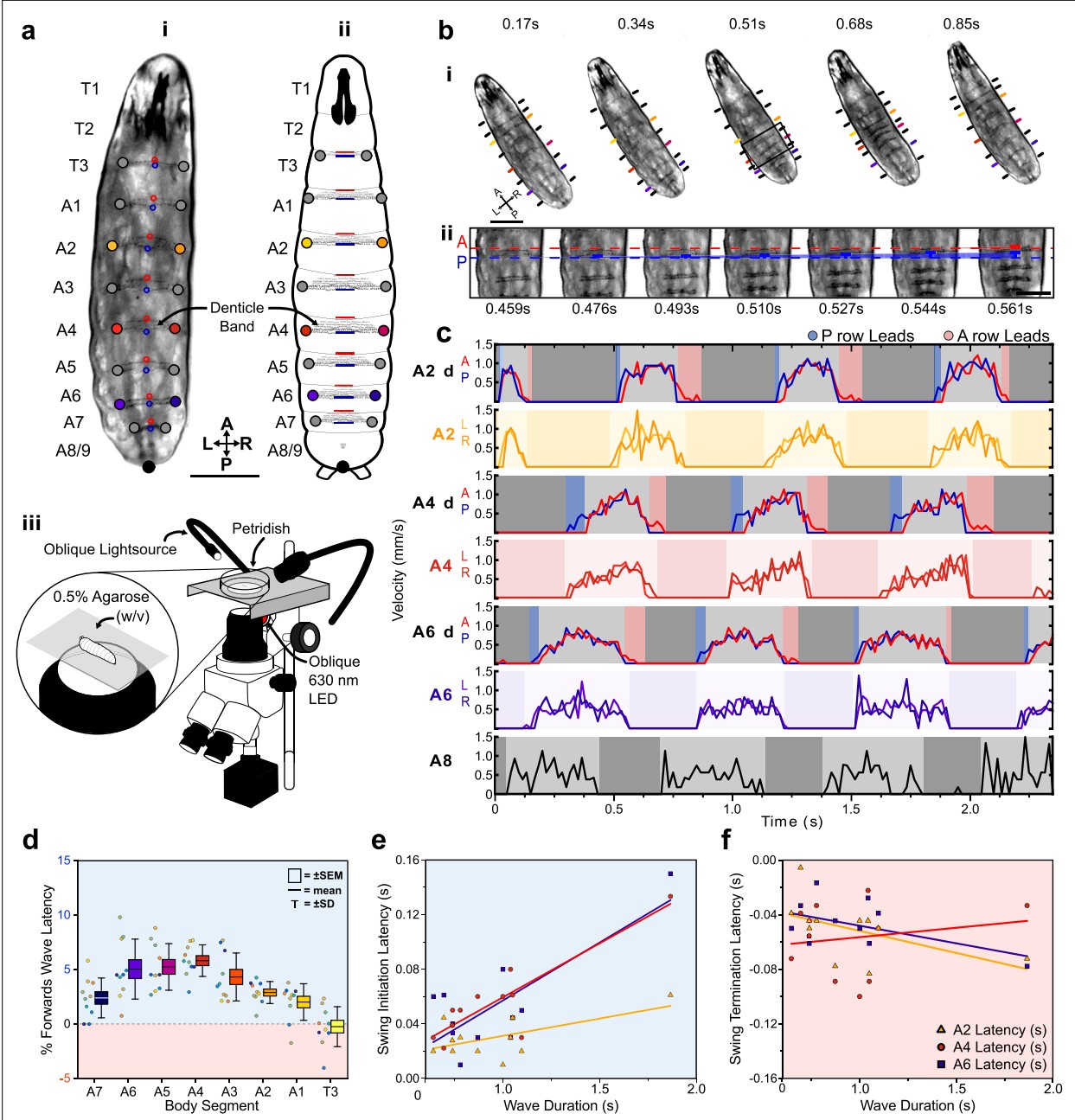

**Figure 2.** Protopodia kinematics follow 'heel-toe'-like footfall dynamics. (**a**) (**i**) Brightfield image and (**ii**) schematic of second-instar larvae showing ventral side denticle belts which reside upon the protopodia and (**iii**) schematic of the imaging setup used for kinematic tracking. Scale bar = 200 μm. (**b**) (**i**) As a forward wave travels through the animal, the distance between denticle bands decreases. Scale bar = 200 μm. (**ii**) At higher frame rate and magnification, changes in distance between the posterior and anterior-most denticle rows are resolved. The posterior-most row (P, blue) initiates movement first and moves until nearly reaching the anterior-most row (A, red) at 0.544 s, after which point, they move together (0.561 s). Scale bar = 100 μm. (**c**) Velocity of anterior- and posterior-most denticles rows (A2d A/P, A4d A/P, A6d A/P) and the left/right end of denticle bands (A2 L/R, A4 L/R, A6 L/R, and A8 L/R) over three representative forward waves, showing how the strategy observed in (**b**) is maintained across body segments. Background colours indicate swing initiation (SI, blue), swing period (SwP, light grey), swing termination (ST, pink), and stance period (StP, dark grey). (**d**) Forward wave latency for different animals and body segments. Positive values denote posterior row led latency. n = 10 animals, 30 waves. (**e**) SI latency scales with wave duration in the posterior abdomen (A6: $R^2$ = 0.61, purple; A4: $R^2$ = 0.78, red) but less so for the anterior abdomen (A2: $R^2$ = 0.35, yellow). n = 12 animals with three latency periods per segment. (**f**) ST latencies do not scale with wave duration (A6: $R^2$ = 0.26, A4: $R^2$ = 0.26, A2: $R^2$ = 0.03). n = 12 animals with three latency periods per segment.

The online version of this article includes the following figure supplement(s) for figure 2:

**Figure supplement 1.** Backward waves show a reversed heel toe rule.

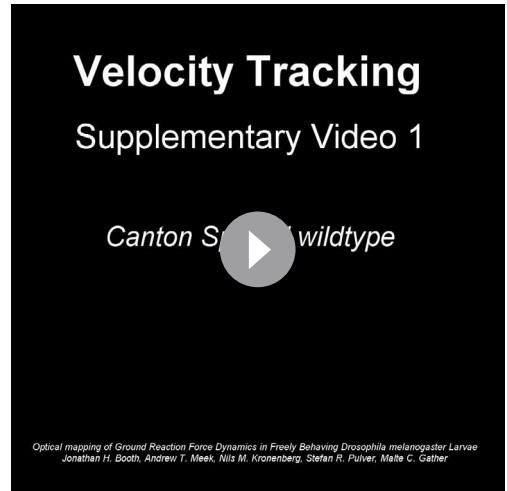

**Video 2.** Kinematic tracking of forward and backward peristaltic waves. Manual tracking of 33 points across the body during forward and backward peristalses.
https://elifesciences.org/articles/87746/figures#video2

detachment in A7. The mean delay was 0.66 s ± 0.21 s (*Figure 4h*, n = 20 waves across six animals).

Next, to estimate the GRF associated with the indentation of each protopodium, we integrated the displacement and stress maps over the region covered by each protopodium. During forward waves, the temporal evolution of GRFs mirrored the characteristics of the cycle seen in the stress maps, with absolute GRFs ranging between 1 and 7 µN (*Figure 5A*). However, unexpectedly, we observed an additional force applied to the substrate both when protopodia leave the substrate (SI) and when they are replanted (ST). To investigate whether this force was due to an active behaviour or due to shifting body mass, we plotted protopodial GRFs against the contact area for each protopodium over time, combining data from multiple forwards waves (*Figure 5B*). We found that the magnitude of force output was positively correlated with protopodial contact area in a quadratic relationship (A6: Adj. $R^2$ = 0.77, A4: Adj. $R^2$ = 0.92, A2: Adj. $R^2$ = 0.79) Comparing different animals, we find that GRFs were relatively consistent across most segments (*Figure 5C*).

The contact area of each protopodium showed a pronounced peak during SI and ST. The maximum contact area during ST was significantly greater than that during SI for the posterior abdomen (p≤0.05 for A8/9-A3) but not for the anterior abdominal protopodium (p>0.05 for A2) (*Figure 5D*). The peak of the displaced volumes during SI was largely determined by wave duration ($R^2$ range: 0.48–0.69, A7-A4, *Figure 5E*), again except for the anterior abdomen (A3: $R^2$ = 0.15; A2: $R^2$ = 0.24). However, the peak of the displaced volumes during ST did not scale with wave duration ($R^2$ range: 0.03–0.05, A7-A2). This suggests that protopodia push off from the substrate harder during faster waves, but that varying wave speed does not strongly influence forces exerted onto the substrate during protopodia placement. This observation is consistent with our morphometric data, which showed that wave duration is associated with SI latencies but not with ST latencies.

## Sub-protopodial force dynamics

Lastly, to investigate how forces are translated into the substrate within a single protopodium during a 'footfall' cycle, we examined the spatiotemporal substrate interaction during the ST (*Figure 6a*). This showed how protopodia expand their indentive contact across both the AP and mediolateral (ML) axes when being replanted. Kymographs along the AP midline of animals and profiles running up the AP axis extracted from these revealed a delay between when the most posterior and the most anterior part of the protopodium contacts the substrate (*Figure 6b*). The mean contact delay relative to the most posterior part of the protopodium was 0.035 s ± 0.007 s at 6 µm away from the most posterior part and increased to 0.062 s ± 0.021 s and 0.253 ± 0.115 s in the middle and at the most anterior part of the protopodium, respectively (*Figure 6c*).

To examine how protopodia expand along the ML axis, we performed a similar analysis, taking kymographs and profiles for the displacement maps at different distances to the midline of a protopodium. At a medial distance from the midline, the contact delay relative to the midline was 0.045 s ± 0.022 s (left) and 0.057 s ± 0.019 s (right). At the distal left and right of the protopodium, contact occurred 0.111 s ± 0.030 s (left) and 0.165 s ± 0.058 s (right) after midline contact (*Figure 6d*). This analysis also showed that protopodia insert a medial-spike into the substrate, through which ST related peak GRFs are conferred, before expanding along the AP and ML axes.

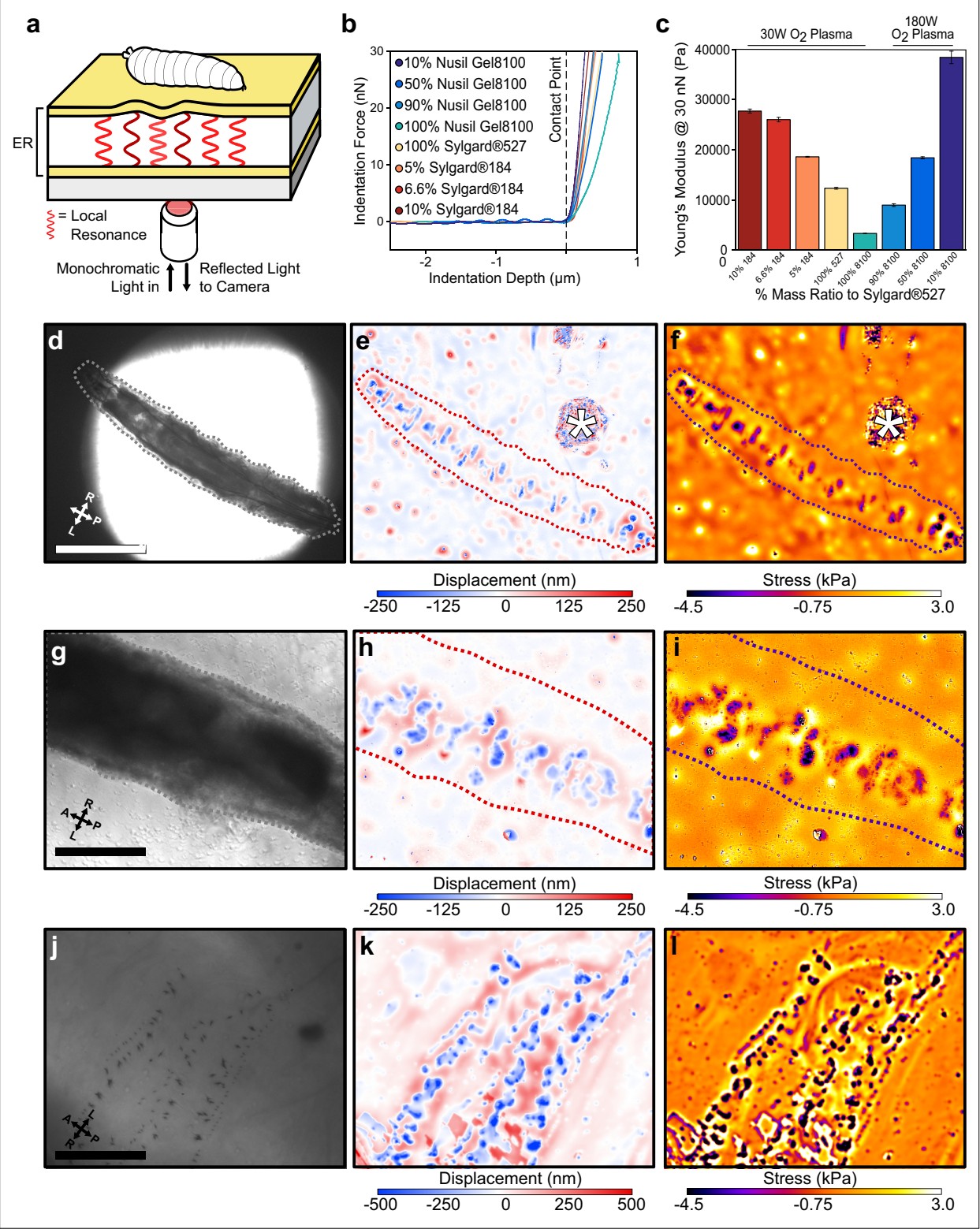

**Figure 3.** Elastic resonator interference stress microscopy (ERISM) maps mechanical substrate interactions in *Drosophila* larvae. (**a**) Schematic of setup for ERISM with *Drosophila* larva on an optical microcavity. Maps of local cavity deformation (displacement) due to indentation forces are generated by analysing cavity resonances. (**b**) Force distance relationship measured by atomic force microscopy (AFM) and (**c**) mechanical stiffnesses (Young's moduli) for microcavities produced by mixing different elastomers at different ratios and applying different plasma conditions. (**d, g, j**) Brightfield images of anaesthetised second-instar larvae recorded at low, medium, and high magnification. (**e, h, k**) Corresponding maps of microcavity displacement. (*

*Figure 3 continued on next page*

*Figure 3 continued*

denotes contamination on cavity surface from handling the larva.) (**f, i, l**) Corresponding maps of mechanical stress obtained by finite element analysis of displacement maps, showing the stress on the substrate due to passive interaction between larvae and substrate. Scale bar = 500 µm (**d**), 250 µm (**g**), and 50 µm (**j**). Images representative of four separate second-instar larvae. Microcavities in (**d–i**) used 30 W $O_2$ 10% Sylgard184 design, and (**j–l**) used a 30 W $O_2$ 5% Sylgard184 design.

The online version of this article includes the following figure supplement(s) for figure 3:

**Figure supplement 1.** Optical setup for elastic resonator interference stress microscopy (ERISM) and wavelength alternating resonance pressure (WARP) experiments.

**Figure supplement 2.** Confirmation of finite element method (FEM) simulation accuracy.

## Discussion

### *Drosophila* larvae, though legless, have protopodia

The cuticle of larvae shows distinct patterns of denticulation (denticle bands) and the developmental processes which give rise to these features have been well studied (*Payre, 2004*), though their role in locomotion has long been unclear (*Fitzpatrick and Szewczyk, 2005*). Here, we show that denticle bands are situated upon larger articulated foot-like cuticular processes, which act as locomotory appendages. Protopodia dynamically change shape during locomotion, allowing sequestration and presentation of denticles. Individual protopodia and individual denticles exert GRFs in the 1–7 µN and 1–48 nN ranges, respectively. Superficially, protopodia resemble the much smaller pseudopodia in cells – transient structures, similarly covered with actin protrusions, used by cells to facilitate movement (*Burnette et al., 2014*). The same function and principles of protopodia may underlie 'creeping welts' noted in larger dipteran larvae (*Friesen et al., 2015*) and show similarities to soft prolegs of *Manduca sexta* caterpillars but are approximately 30 times smaller (*Lin and Trimmer, 2010*).

### Insights from morphometric kinematic tracking of denticle band movements

Our study provides, to our knowledge, the first detailed description of the morphometry of denticle bands during movement, showing how denticle bands are deployed onto and removed from the substrate. Posterior denticle rows hit the substrate before anterior rows during deployment (ST) and left the substrate before anterior rows during removal (SI). This suggests that both deployment and removal involved rolling 'heal-toe' like movements, similar to footfalls in limbed animals, including terrestrial arthropods (*Federle and Labonte, 2019*). Removal but not deployment correlated with wave duration. In *Manduca* caterpillars, it has been noted that SwPs scale positively with wave duration (*Simon et al., 2010*); however, to our knowledge, there is no measurement for SI and ST in these animals.

SI latencies scaled positively with wave duration across most segments whereas ST latencies did not show this trend. SIs scale with SwP, and this could be mediated by proprioceptor activity in the periphery (*Vaadia et al., 2019*). Fine sensorimotor control of musculature during this process would allow for precisely tuned propulsion during peristalsis. In contrast, the more random nature of the ST suggests the process is less finely controlled. This could be a consequence of fluid inertia within the animal and/or the release of elastic energy from cuticle (*Sun et al., 2022*) or relaxation of muscles (*Simon et al., 2010*; *Ormerod et al., 2022*).

### ERISM-WARP allows computation of GRFs in *Drosophila* larvae

We adapted state-of-the-art mechanobiological force measuring techniques to enable measurement of substrate interaction dynamics of a freely behaving soft-bodied animal with micrometre spatial resolution, millisecond temporal resolution, and nanonewton force resolution. Previously, high-resolution force mapping was limited to cellular mechanobiology. Specifically, we developed microcavity resonators tuned to the vertical forces generated by larvae and employed ERISM and WARP to perform direct measurements of substrate interactions in anaesthetised and behaving animals. GRFs produced by individual denticles in anaesthetised animals were in the ~11 nN range. The measured vertical GRFs produced by the individual protopodia of each segment were in the 1–7 µN range, roughly three orders of magnitude less than the 17 mN recorded from an entire 1.72 g *M. sexta* caterpillar (*Lin and*

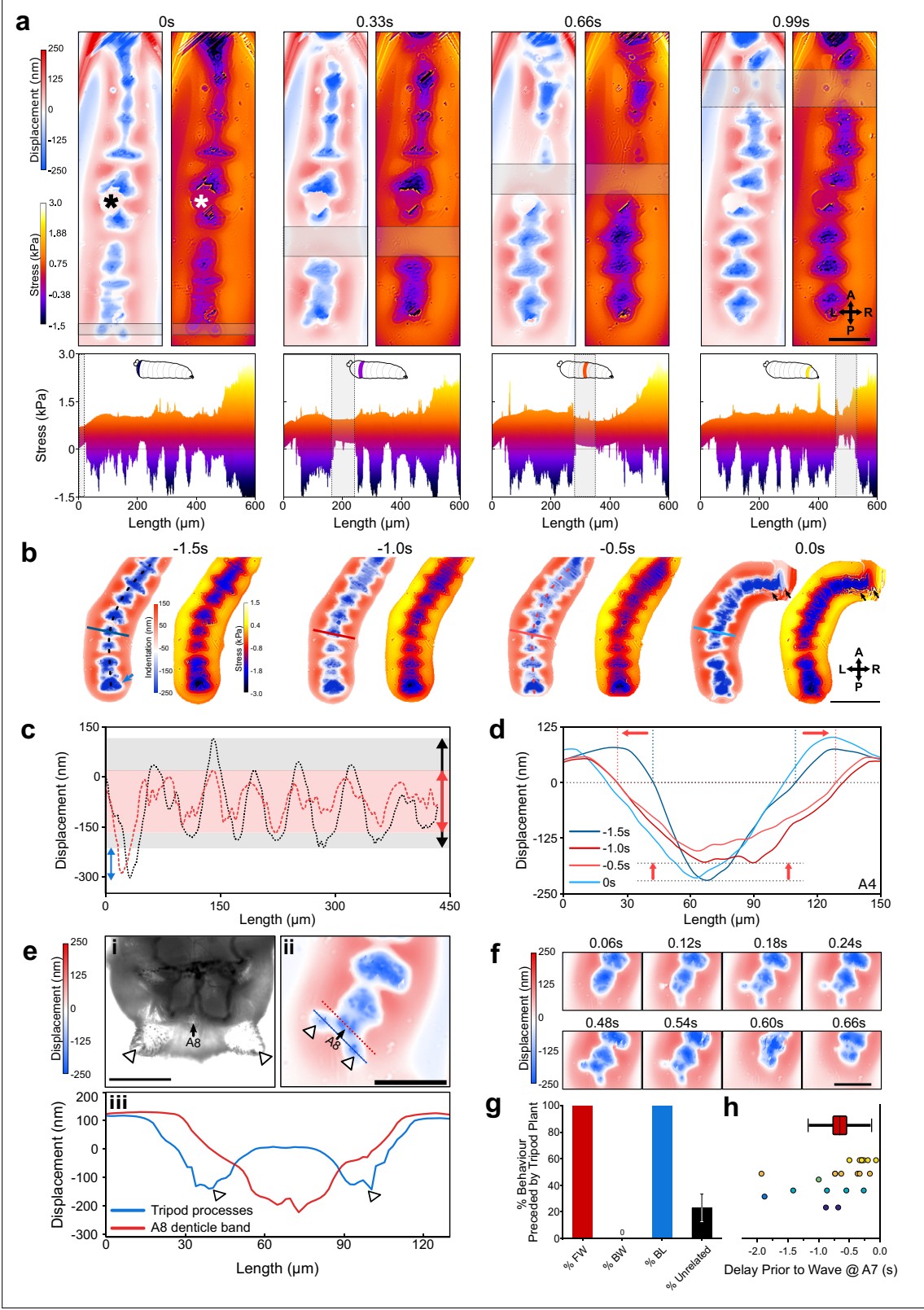

**Figure 4.** Wavelength alternating resonance pressure (WARP) imaging reveals dynamics of substrate interactions during larval movement. (**a**) WARP image sequence of displacement and stress maps (top) for a freely behaving second-instar larva during forward locomotion. (* denotes dust artefact.) Lateral projections of stress maps (bottom) showing individual protopodia interdigitated by naked cuticle. As a contractile wave (grey box) progressed through the animal, protopodia were lifted off the substrate. Scale bar = 100 μm. (**b**) WARP image sequence of larva prior to (–1.5 s to –0.5 s) and

*Figure 4 continued on next page*

*Figure 4 continued*

engaging in (0 s) a headsweep (representative of two animals and three turns). Note the large posterior displacement (blue arrow; images cropped around the animal). Scale bar = 200 µm. (**c**) Profiles of cavity displacement along anteroposterior (A-P) axis in resting state (black dotted line at –1.5 s in **b**) and pre-headsweep (red dotted line at –0.5 s in **b**), showing that peak displacement decreased across all segments from the resting state (grey box) to pre-headsweep (pink box). (**d**) Bilateral displacement profile across the mediolateral (ML) axis of the A4 protopodium (solid lines in **b**) at different times prior to the headsweep, showing that the width of the contact increases from the resting state (–1.5 s) to the pre-headsweep state (–0.5 s) and partially reduces again immediately after head movement. (**e, i**) Brightfield image (third-instar larva) and (**ii**) displacement map (second-instar larva) of the posterior-most body segment, showing how two cuticular protrusions (white arrowheads) and the terminal protopodium (A8) generate a tripod-shaped substrate displacement. (**iii**) Profiles along blue and red dotted lines in (**ii**). Scale bar = 200 µm (**i**) and 100 µm (**ii**). (**f**) Sequence of displacement maps of tripod structure before the start of a forward wave (<0.24 s) and the removal of tripods upon beginning of peristalsis (>0.48 s). Scale bar = 100 µm. (**g**) Percentage of forward waves (FW), bilateralisms (BL), backward waves (BW) preceded by tripod contact, and tripod deployments without any observed locomotor behaviour (unrelated). (**h**) Time delay between tripod deployment and initiation of movement at A7. Points colour-coded by animal, n = 6. Line = mean, box = ±1 standard error of the mean, whiskers = ±1 standard deviation.

The online version of this article includes the following figure supplement(s) for figure 4:

**Figure supplement 1.** Ordinary larval behaviour is maintained on collagen-treated microcavities compared to commonly used agarose substrates.

**Figure supplement 2.** Wavelength alternating resonance pressure (WARP) computation pipeline.

*Trimmer, 2010*). Our measurements provide fundamental constraints for future biomechanical modelling studies seeking to incorporate these structures.

Displacement and stress maps produced during larval crawling revealed that animals can control when and how protopodia contact the substrate. We observed that larvae travel surrounded by moisture from a water droplet, which produces a relatively large upwardly directed force in a ring around the animal. This surface tension produced by such a water droplet likely serves a role in adhering the animal to the substrate. However, during forward waves, we found that protopodia detached completely during SwP, suggesting this surface tension-related adhesion force can be easily overcome by the behaving animal. This observation, coupled with our lateral imaging of protopodia in constrained animals, explains how larvae prevent their rough denticulated cuticle from creating drag due to friction against the direction of the wave. Larvae do not simply pull protopodia off the substrate in a vertical direction; instead, they horizontally slide posterior regions forward in the axis of travel, before invaginating and therefore sequestering friction-generating features (e.g. denticles). This shows similarities to the use of shearing forces to detach adhesive pads in limbed arthropods (*Federle and Labonte, 2019*). Inversion of the

**Video 3.** Wavelength alternating resonance pressure (WARP) imaging during forward peristalses. Video showing high frame rate displacement maps produced by a freely behaving *Drosophila* larva. Displacement maps were high-pass Fourier filtered to make denticulated cuticle more readily visible and projected in 3D to show the effects of substrate interaction. Details of the Fourier filtering procedure are described in a previous study (*Kronenberg et al., 2017b*).
https://elifesciences.org/articles/87746/figures#video3

**Video 4.** Interference mapping of body mass redistribution during anterior bilateral behaviours. Video showing the raw reflection data during the preparatory phase of bilateral behaviours.
https://elifesciences.org/articles/87746/figures#video4

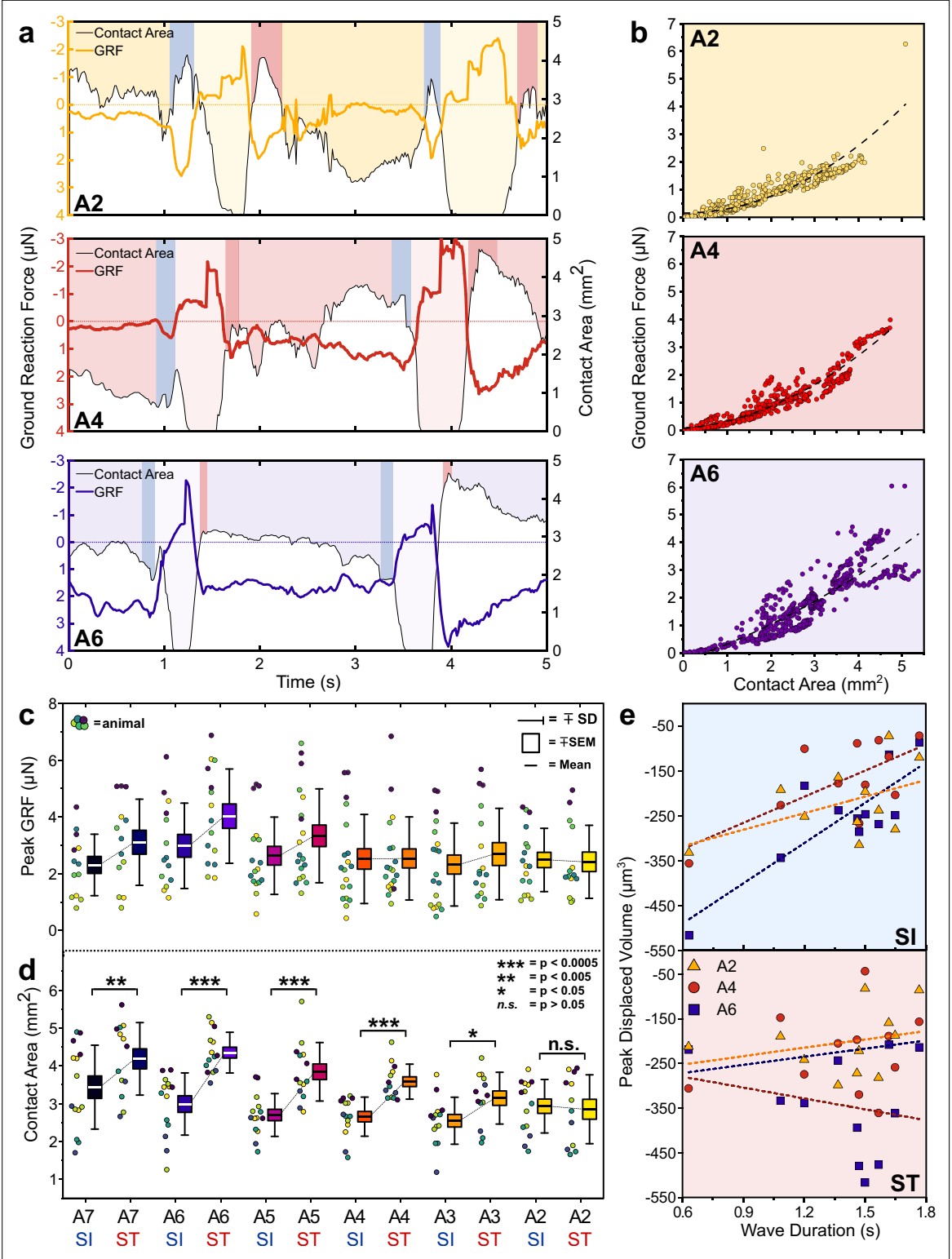

**Figure 5.** Protopodia produce ground reaction forces (GRFs) in the micronewton range and show complex spatiotemporal dynamics. (**a**) GRF (coloured line) and protopodial contact areas (white area under black line) during forward crawling for A2, A4, and A6 protopodia, showing progression of waves through animal (light-coloured boxes). Blue (SI) and pink (ST) boxes denote characteristic troughs in GRF immediately prior to protopodia leaving the substrate and returning to the substrate, respectively. (**b**) GRFs exerted by different protopodia show a second-order polynomial relationship (dashed line) with the contact area of that protopodium (A6: Adj. R² = 0.77, A4: Adj. R² = 0.92, A2: Adj. R² = 0.79). (**c**) Peak GRFs and (**d**) peak contact area during SI and ST across body segments. Data points denote single events, colours indicate different animals. n = 5, 15 waves. Contact areas were compared

*Figure 5 continued on next page*

Figure 5 continued

by a two-way repeated-measures ANOVA (*<0.05, **<0.005, ***<0.0005, n.s. = not significant). (**e**) During SI, peak displaced volume scaled with wave duration for larger abdominal segments (A6: $R^2 = 0.69$; A4: $R^2 = 0.48$) but not for smaller anterior segments (A2: $R^2 = 0.24$). During ST, displaced volume did not scale with wave duration regardless of the segment (A6: $R^2 = 0.05$; A4: $R^2 = 0.05$; A2: $R^2 = 0.08$). n = 4, 11 waves.

cuticle to remove denticles from the substrate may also explain why natural variations in denticle count across animals do not strongly affect locomotor behaviour (*Fitzpatrick and Szewczyk, 2005*). The invagination process is reversed in order to expand the protopodia into and locally across the substrate, providing an expanding anchor which can serve as a postural support to enable locomotion and prevent lateral rolling during bilaterally asymmetric behaviours such as headsweeps. The dynamic anchoring during the progression of peristaltic waves thus serves to counteract horizontal reaction forces resulting from Newton's third law of motion. Such a sequence of positioning points of support and anchoring them against the substrate has long been postulated to be a fundamental process in soft-bodied locomotor systems (*Trueman, 1975*) and may be central to explaining why soft-bodied animals have evolved segmentally repeating bodies (*Budd, 2001*). However, WARP microscopy is largely limited to measurements of forces in the vertical direction, and though we can make inferences such as this as they are a consequence of fundamental laws of physics, we present this conclusion as a testable prediction which could be confirmed using a force measurement technique more tuned to horizontally directed forces relative to the substrate.

Our ERISM-WARP measurements also revealed substrate interaction from accessory structures. Immediately before enacting headsweep, larvae redistributed their body mass into naked cuticle in between protopodia along the midline, effectively fusing multiple protopodia into a single 'ultra-protopodia' that extends across multiple posterior segments. This redistribution occurs hundreds of milliseconds before the start of a headsweep, suggesting that it may be part of an active preparatory behaviour. Similar preparatory behaviours have been observed in caterpillars before cantilevering behaviours (*Lin et al., 2011*), adult flies during fast escape behaviours (*Card and Dickinson, 2008*), and humans during stepping (*Watanabe and Higuchi, 2022*). More detailed characterisation of this behaviour remains a challenge owing to the changing position of the mouth hooks. Due to their rigid structure and the relatively large forces produced in planting, mouth hooks produce substrate interaction patterns which our technique struggles to map accurately due to overlapping interference fringes ambiguating the fringe transitions.

We also observed transient tripod-shaped substrate interactions in posterior terminal regions of larvae immediately before forward waves and headsweeps. Two bilateral cuticular protrusions covered in trichomes, labelled in previous work as anal papillae (*Zanini et al., 2016*), are likely candidates responsible for these substrate interactions. However, the actions of these structures have hitherto not previously been described as a part of movement in soft-bodied animals. Each body segment has a preceding substrate-planted segment which acts as the anchor and lever to push the animal forward. However, A8 is an exception; it has no full preceding segment in contact with the substrate to counteract its muscle contraction. The tripod processes are ideally positioned to provide an anchor against horizontal reaction force generated by the initial contraction when moving forward (*Figure 7a*) and might effectively form a temporary extra segment prior to initiation of a wave (*Figure 7b*). The deployment of cuticular features as transient anchors has not been a focus of previous studies; future work should incorporate our findings into models of crawling behaviour. WARP and ERISM have technical limitations, such as the difficulty of resonator fabrication. This problem is compounded by the fragility of the devices owing to the fragility of the thin gold-top mirror. This becomes problematic when placing animals onto the microcavities as often the area local to the initial placement of the animal is damaged by the paintbrush used to move the animals. Further, as a result of the combining of the two wavelengths, the effective frame rate of the resultant displacement and stress maps is equal to half of the recorded frame rate of the interference maps. This necessitates recording at very high frame rates and thus requires imaging at reduced image size to maximise frame rates, but this in turn reduces the number of peristaltic waves recorded before the animal escapes the field of view. A further limitation is that WARP and ERISM are sensitive mainly to forces in the vertical direction; this is complementary to TFM, which is sensitive to forces in horizontal directions. Using WARP in conjunction with high-speed TFM (possibly using tuneable elastomers presented here) could provide a fully integrated picture of underlying vertical and horizontal traction forces during larval locomotion.

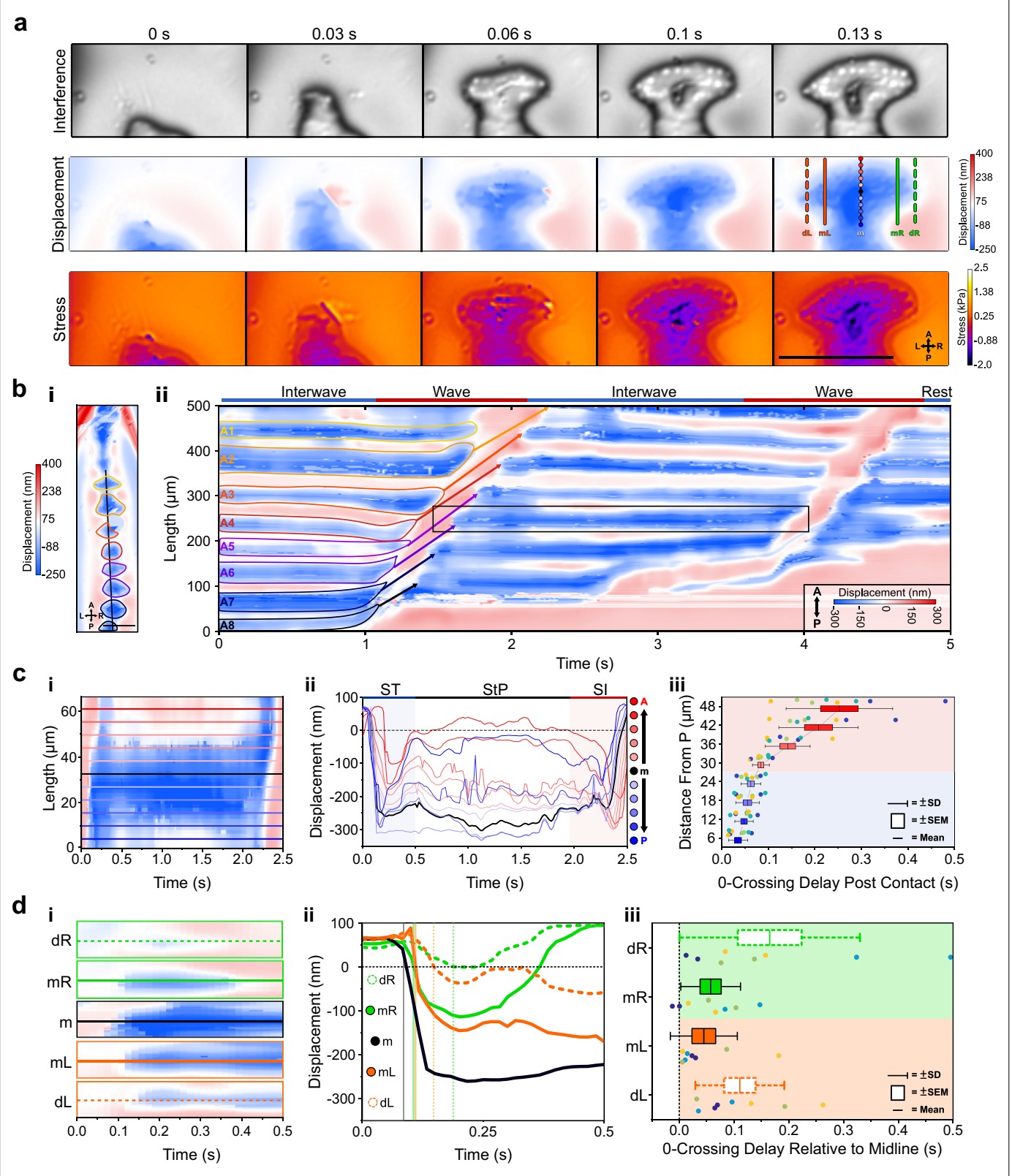

**Figure 6.** Sub-protopodial force dynamics reveal sub-step processes and functional substrate-interfacing domains in each step. (**a**) Wavelength alternating resonance pressure (WARP) imaging of protopodial landing during swing termination (ST) of an A6 protopodium. Raw interference images from WARP acquisition show footprints of individual denticles as white dots. Displacement and stress maps show how landing starts with posterior denticle rows before spreading out along the anteroposterior (AP) and mediolateral (ML) axes. Scale bar = 100 μm. (**b, i**) Displacement map of whole

*Figure 6 continued on next page*

*Figure 6 continued*

animal. (**ii**) Kymograph of displacement along AP axis (black line in **i**) over two forward waves. Bands of red and blue correspond to naked cuticle and protopodia, respectively. Scale bar = 100 µm. (**c, i**) Kymograph of displacement along the AP axis of an A6 protopodium (box in **b**). (**ii**) Profiles across kymograph at different positions along the AP axis of protopodium (lines in **i**). (**iii**) Latency of substrate indentation (displacement <0 nm) during ST along the AP axis, relative to the extreme posterior of protopodium. Compared to the posterior half of protopodium (light blue area), the anterior half shows larger latencies and variations in latency (light red area). n = 4 animals, eight ST events. (**d**) Kymograph of displacement along AP axis during ST for the distal left (dL), medial left (mL), midline (m), medial right (mR), and distal right (dR) section of the A6 protopodium. Height of each kymograph, 66.42 µm. (**ii**) Profiles across the central AP line of each kymograph in (**i**). Vertical lines indicate times when midline, medial right/left, and distal right/left indentation starts (displacement <0 nm). (**iii**) Latency of substrate indentation during ST relative to the midline for medial right/left and distal right/left locations. n = 4, eight swing termination events.

## Evidence for functional subdivisions within protopodia

By examining the dynamics of individual footfalls, we found that protopodia exhibited characteristic spatiotemporal force patterns across the footfall cycle. This shows parallels to the regional specificity of function in a vertebrate foot. Specifically, the posterior medial region of the protopodia makes a large contribution to peak GRFs exerted during ST (*Figure 7b*), similar in nature to a vertebrate heel strike impacting the surface prior to the rest of the foot. We propose that this zone of the protopodia acts as a vaulting point for the protopodia, functioning as a 'point d'appui' (point of support) as proposed in other soft-bodied animals (*Trueman, 1975*; *Valentine, 1989*). The transience of this vaulting point suggests it may be critical for locomotion, but dispensable for postural control during StP. The distal area of protopodia exhibited a similar transience. This increased force transmitted into the substrate is unexpected as the forces generated for the initiation of movement should arise from the contraction of the somatic muscles. We propose that the contraction of the musculature responsible for sequestration acts to move haemolymph into the protopodia, thus exerting an increased pressure onto the substrate while the contact area decreases as a consequence of the initiation of sequestration. Immediately after the posterior and medial protopodia impact during ST, the contact area of the outer region of the protopodia grew across both the AP and the ML axes. However, throughout the StP, this outer region then slowly retracted, suggesting it too was not critically important for maintaining posture during StP. This may reflect a transient anchoring mechanism – specifically, this anchor region deploys to provide greater friction for the subsequent segments (*Figure 7c*). This would allow the contractile wave to progress unimpeded by resultant reaction forces. Previously, such a function was thought to be provided mainly by mucoid adhesion (*Trueman, 1975*). However, *Drosophila* larvae are proficient at crawling over wet surfaces where mucoid adhesion is reduced or impossible (*Apostolopoulou et al., 2014*). Larvae can adhere to dry surfaces but have difficulty moving over these, although mucoid adhesion would provide optimal anchorage in this context. Water surface films appear to facilitate larval locomotion in general, but the biomechanical mechanisms by which this occurs remain unclear. We propose that protopodia act to provide an optimal balance between anchorage and adhesion depending on the environmental context. Overall, our work suggests that *Drosophila* larvae use a sophisticated process of articulating, positioning, and sequestering protopodia to enable movement over terrain. Future work will be needed to determine the extent to which these processes are conserved across other soft-bodied crawlers.

## Conclusions and outlook for future work

Combining ERISM-WARP with a genetically tractable model organism opens new avenues for understanding the biomechanical basis of animal behaviour, as well as the operation of miniaturised machines. Here we have provided new insights into the relatively well-studied behaviour of *Drosophila* larval locomotion. We have provided new quantitative details regarding the GRFs produced by locomoting larvae with high spatiotemporal resolution. This mapping allowed the first detailed observations of how these animals mitigate friction at the substrate interface and thus provide new insights into how locomotion is achieved in soft animals. Further, we have ascribed new locomotor function to appendages not previously implicated in locomotion in the form of tripod papillae, providing a new working hypothesis for how these animals initiate movement. It is our hope that these new principles underlying locomotion outlined here serve as useful biomechanical constraints as called for by the wider modelling community (*Tytell et al., 2011*). We used *Drosophila* larvae as a test case, but our methods now allow elastic optical resonators to be tuned to a wide range of animal sizes and thus

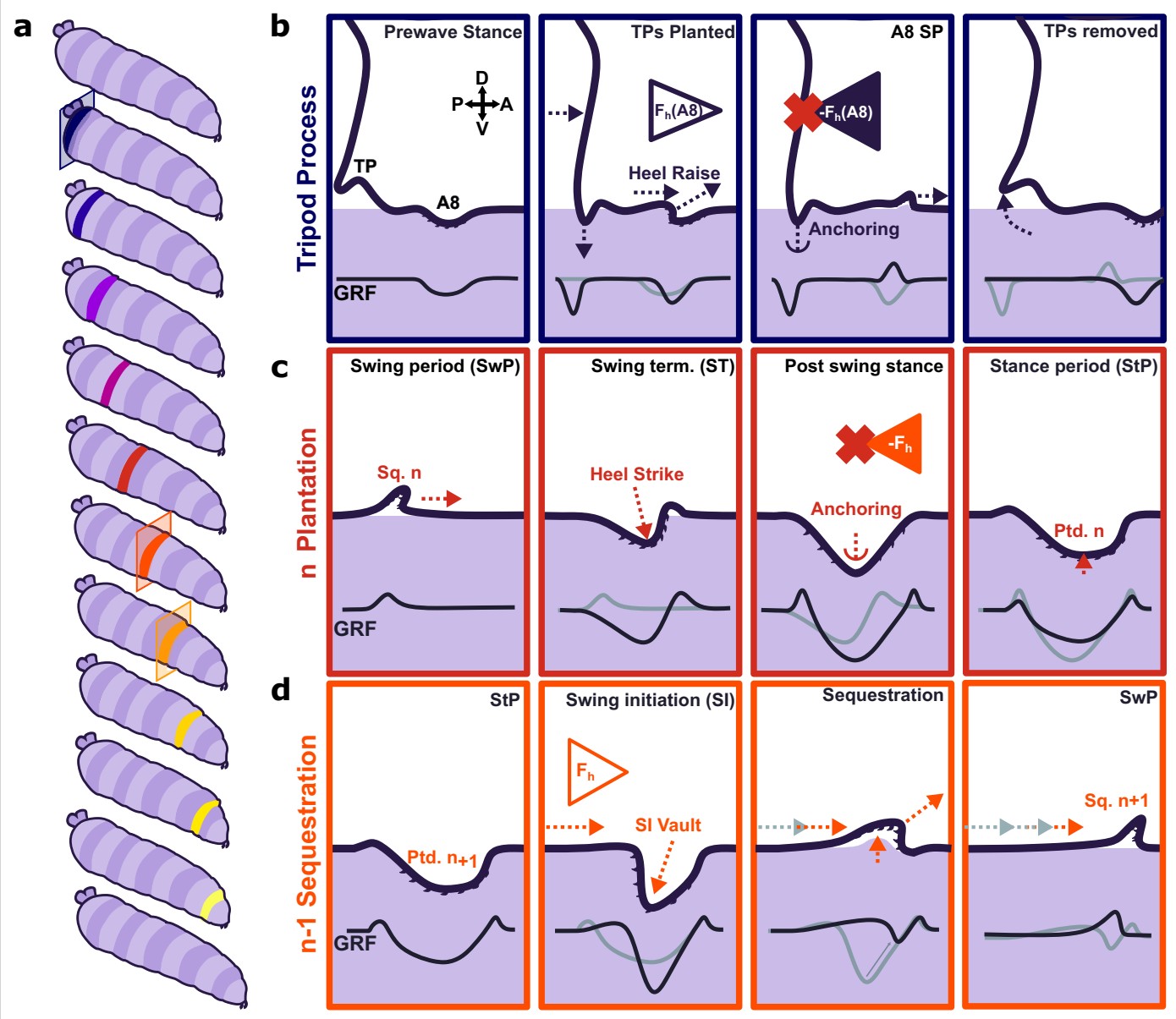

**Figure 7.** Proposed model for protopodia–substrate interactions during *Drosophila* larval locomotion. (**a**) Schematic illustration of forward wave propagating from posterior (blue) to anterior (yellow). (**b**) At the start of a forward wave, animals contract the posterior-most abdominal segment (A8), producing an anterograde horizontal force $F_h$ (A8). Due to Newton's third law, there is an equal but opposite reaction force $-F_h$ (A8). To counteract this force, tripod processes (TPs) deploy onto the substrate and generate a temporary anchor, allowing the A8 protopodium to swing forward. (**c**) During swing termination (ST) at the end of the swing period (SwP) of segment *n*, the corresponding sequestered protopodium (Sq. n) strikes the substrate with its posterior most denticle row, then gradually unfolds into the substrate along its entire anteroposterior extent. During the stance period (StP), this planted segment *n* (Ptd. n) forms an anchor to mitigate the retrograde reaction force due to the subsequent contraction of segment *n*-1. (**d**) In time with anchoring of protopodium *n*, protopodium *n*-1 performs swing initiation (SI) by removing denticles from the substrate and sequestering into an invagination pocket, which reduces friction during the subsequent SwP. The contraction of segment *n*-1 then leads to an anterograde force ($F_h$) that is balanced by the anchoring of protopodium *n* as illustrated in (**c**).

create new possibilities for studying principles of neuro-biomechanics across an array of animals. In parallel, roboticists are increasingly moving to create miniaturised soft robots for a variety of applications. Our approach is well suited to provide ground truth, constraints, and inspiration for the development of such miniaturised machines. It also provides a potentially powerful new resource for evaluating the performance of these devices as our methodology will also allow scientists to measure GRFs during the operation of miniaturised soft machines. Importantly, while we have focused here

on the movement of soft animals, our sensors could also be tuned to measure forces produced by small-limbed animals or miniaturised machines with rigid internal or external skeletons. Overall, this work therefore establishes a flexible platform for future investigations aimed at integrating knowledge across genetics, neuroethology, biomechanics, and robotics.

# Materials and methods

## Key resources table

| Reagent type (species) or resource | Designation | Source or reference | Identifiers | Additional information |
|---|---|---|---|---|
| Biological sample (*Drosophila melanogaster*) | Canton S (wildtype) | Bloomingtons Stock Center | FBsn0000274 ID 64349 | |
| Software, algorithm | OriginPro 2019b | OriginLab Corporation | | Statistical analysis and plotting |
| Software, algorithm | COMSOL Multiphysics | COMSOL Inc | | Finite element method simulation resolving stress maps |
| Software, algorithm | Python 3.0 and 2.0 | Anaconda Inc | | Cavity length map computation |
| Software, algorithm | Inkscape v.1.01 | Inkscape Organisation | | Vector figure making |
| Software, algorithm | FIJI | National Institutes of Health/SciJava | 1.52p | Image analysis and manual tracking |
| Chemical compound, drug | Phosphate-buffered saline | Gibco | 10010023 | Collagen coating |
| Chemical compound, drug | Hydrochloric acid 5 M | Sigma-Aldrich | 10605882 | Collagen coating |
| Chemical compound, drug | Acetic acid 95% | VWR | 84528.290 | Collagen coating |
| Chemical compound, drug | Collagen-I | Millipore | L7220 | Collagen coating |
| Other | Gold grains (99.99%) | Kurt J. Lesker Company | EVMAU40SHOT | Microcavity fabrication |
| Other | Chromium 99.95% | Kurt J. Lesker Company | EVMCR35 EJTCRXX351 | Microcavity fabrication |
| Other | Silicon Dioxide Fused quartz target | Kurt J. Lesker Company | EJUSIO2451 | Microcavity fabrication |
| Chemical compound, drug | NusilGel8100 | Nusil | GEL-8100 | Microcavity fabrication |
| Chemical compound, drug | Sygard527 | Dowsil | 2270030 | Microcavity fabrication |
| Chemical compound, drug | Sylgard184 | Dowsil | 1673921 | Microcavity fabrication |
| Chemical compound, drug | Ethyl butanoate | VWR | ACRO118182500 | Retaining animals within field of view |
| Chemical compound, drug | Mineral (Paraffin) oil | VWR | 31911.D9 | Suspension of ethyl butanoate |
| Other | 24 mm² glass substrate | ORSAtec | 2.01.03.0167.59.16.1 | Microcavity fabrication |
| Other | FlexAFM | Nanosurf | | Atomic force microscope |
| Other | uniqprobe Cantilevers | Nanosensors | qp-CONT | Stiffness calibration by atomic force microscopy |

*Continued on next page*

*Continued*

| Reagent type (species) or resource | Designation | Source or reference | Identifiers | Additional information |
|---|---|---|---|---|
| Other | CM110 Monochromator | Spectral Products | | Monochromator for scanning wavelength ERISM |
| Other | Optical cage system components | Thorlabs | | Cage system for ERISM and WARP, see supplementary information |
| Other | EMS 6000 Photoresist Spincoater | Electronic Microsystems | EMS 6000 | Microcavity fabrication |
| Other | Ultra high vacuum deposition chamber | Ångstrom Engineering | | Microcavity fabrication |
| Other | Andor Zyla 4.2 10-Tap | Andor Technology | | WARP and ERISM image acquisition |
| Other | iCube CMOS | NET GmbH | NS4203BU | Brightfield image acquisition |
| Other | XIMEA CMOS | XIMEA GmbH | MQ013MG-E2 | Behavioural image acquisition |
| Chemical compound, drug | CHAPS, 3-[(3-cholamidopropyl)dimethyl ammonio]–1-propane sulfonate | Acros Organics | 10834531 | Electrostatic buffer for atomic force microscopy measurements |

## Animal rearing

Animals were raised on standard cornmeal and yeast medium (17.4 g/L yeast, 73.1 g/L cornmeal, 5.8 g/L agar, 4.8 ml/L propionate) at 25°C with a 12 hr light-dark cycle except where explicitly stated otherwise. Animals were given at least 1 hr to acclimate to room temperature prior to all experiments. *Canton S* (CS) wildtype larvae were used for all experiments (Fly Base Identifier: FBsn0000274). Immediately prior to experiments, samples of media containing larvae were taken using a spatula before being placed into a columnar stacked sieve with 40, 60, and 100 meshes from top to bottom, respectively. Media samples were run under gentle flowing tap water to separate adult debris, second-instar larvae, and first-instar larvae with embryos on each mesh. Larvae from the 60-mesh fraction of the sieve were observed under a microscope, and animals around 1 mm were selected and washed before being placed on 1% (w/v) agarose-lined dishes.

## Microcavity fabrication

The fabrication protocol of elastic microcavities was adapted from *Kronenberg et al., 2017a*. 24 mm$^2$ borosilicate glass substrates of No.5 thickness were cleaned via ultrasonication in acetone followed by propan-2-ol for 3 min. After cleaning, substrates were dried using $N_2$ and baked at 125°C for 10 min to clear any residual solvent. Cleaned glass substrates were then plasma treated with oxygen plasma for 3 min at 20 SCCM $O_2$ flow rate to clear any residual organics and activate the surface of the glass. Cleaned and activated glass substrates were then sputter coated with 0.5 nm of Cr, which acted as an adhesion layer for the subsequent 10 nm Au layer that was deposited by thermal vapour deposition. 50 nm of $SiO_2$ was then deposited by sputter coating to improve stability of the resultant bottom mirrors. Roughly 100 µL of pre-mixed and degassed polydimethylsiloxane gels was spincoated onto the bottom mirrors at 3000 RPM, 1500 RPM acceleration, for 60 s and then quickly transferred to a pre-heated metal plate at 150°C for 1.5 hr to cure the elastomer. After curing, elastomer-coated bottom mirrors were $O_2$ plasma treated with the desired plasma power at 20 SCCM $O_2$ flow rate for 10 s. 15 nm of Au was then deposited onto the oxidised elastomer, thus completing the microcavity.

## Microcavity characterisation

Microcavities were characterised using a NanoSurf Flex Atomic Force Microscope (Nanosurf, Liestal, Switzerland). 15–18-µm-diameter glass beads were glued to the tip of uniqprobe QPCont cantilevers (Nanosensors AG, Neuchatel, Switzerland) using a UV-polymer glue after thermal calibration of the spring constant at 21°C. Sphere-tipped cantilevers were then indented into microcavity samples at 1 µm/s with up to 30 nN of force. This process was repeated across the surface of the microcavity at least five times, with each measurement being roughly 2 mm apart to get a measure of the variation across the cavity surface. Force–distance profiles recorded by the AFM were then fitted to the Hertz

model to compute the Young's modulus at each point of each sample. Mean cavity lengths were measured by taking four ERISM images at ×4 magnification from each corner of the cavity, and then taking the mean of four regions of interest per image.

Prior to use in experiments, a 12-well silicone chamber (ibidi GmbH, Munich, Germany) was cut such that only one large square-well, originally comprised of four smaller wells cut off from the rest of the chamber, remained and was placed onto a microcavity. A low pH Collagen-I (1 mg/ml; Millipore L7220) solution was then prepared at a 1:1 (v/v) ratio with pH3 phosphate-buffered saline (PBS). pH3 PBS was prepared with either hydrochloric acid or acetic acid, mixing until pH3 was recorded using an electronic pH meter. Collagen-I mixtures were then dosed onto microcavities in silicone wells (1 mL per microcavity) and allowed to coat the surface overnight at 4°C. Immediately before the experiment, microcavities were washed with deionised water at least five times, taking care not to remove all liquid to prevent damage to the top gold surface.

## Denticle band kinematic imaging

All animals were raised in ambient light conditions at room temperature. Between 48 and 72 hr after flies were introduced to fresh media, feeding second-instar Canton S wildtype animals were selected with a size-exclusion criterion – any animals below 0.8 mm or above 1.5 mm were rejected. Animals were then washed and allowed to acclimate to 0.5% (w/v) agarose.

Immediately before experiments, a single animal was transferred to a freshly set dish containing 0.5% (w/v) agarose while still transparent. These dishes were then quickly placed onto the 3D-printed stage of a custom-built inverted Bresser Advance ICD stereomicroscope (Bresser GmbH, Rhede, Germany). Denticle band images were acquired, through the still transparent agarose substrate, at 60 frames per second for at least 1 min while the larva was freely behaving. All images were acquired using a XIMEA CMOS camera (XIMEA GmBH, Münster, Germany) through MicroManager 1.4 (*Edelstein et al., 2010*). The velocity of 33 individual identifiable points across the animal's body during peristaltic waves whilst imaging from the ventral side of second-instar larvae Denticle bands were tracked manually using the Manual Tracking plugin of ImageJ (*Schindelin et al., 2012*). Analysis of tracking data was performed using OriginPro 2019 (OriginLab Corporation, MA).

## ERISM and WARP imaging

ERISM was used to record high-resolution maps of substrate indentations by monitoring local changes in the resonances of a soft and deformable optical microcavity. ERISM has been used to quantify cellular forces down to the piconewton range. The static thickness of microcavities was measured adapting our previously published ERISM method as described in *Liehm et al., 2018* and *Kronenberg et al., 2017a*. In brief, images of the cavity were taken under epi-illumination with a series of 201 different wavelengths (550–750 nm in 1 nm steps). From these images, the minima in the spectral reflection for each pixel were correlated with theoretical values obtained from optical modelling for cavities of different thicknesses to determine the actual thickness at each position across the image (cavities were between 8 and 12 μm in static thickness). Thickness maps were converted into maps of local displacement by subtracting a linear plane using the mean thickness of the cavity in each corner.

For dynamic force mapping, we used a further improved version of the WARP routine described in *Meek et al., 2021*. Epi-illumination with light of two different and quickly alternating wavelengths was produced by passing the emission from two identical red LEDs (dominant emission wavelength 625 nm, FWHM 17 nm; Thorlabs Inc NJ) through two identical narrow bandpass filters (peak transmission at 633 nm, FWHM of 1 nm; Thorlabs Inc). By tilting the filter located in front of one of the LEDs by approximately 15° relative to the incident light, its peak transmission wavelength was tuned to $\lambda_\theta$ = 628 nm. For the optical modes supported by our microcavities, this corresponds to a phase shift of roughly 90°, but remains within the same free spectral range band of the cavity. For the WARP measurements, we first took calibration images (under subsequent illumination at $\lambda$ and $\lambda_\theta$) of the empty microcavity in an area with roughly linear slope in cavity thickness, for example, near where the silicone well containing the larvae meets the surface of the cavity. Images of behaving larvae were then recorded under rapidly alternating illumination at $\lambda$ and $\lambda_\theta$, with the camera sending alternating trigger pulses to each LED to generate interleaved stacks of $\lambda$ and $\lambda_\theta$ images.

Displacement maps were obtained from these stacks using a series of image transformations, based around the fact that the ratio of the difference and the sum of pixel intensities at $\lambda$ and $\lambda_\theta$

is linked to local thickness in an unambiguous manner, at least across each free spectral range. See *Figure 4-figure supplement 2* and *Meek et al., 2021* for further details on the calculation of displacement from the $\lambda$ and $\lambda_\theta$ images. All WARP and ERISM images were acquired using an Andor Zyla 4.2 sCMOS camera (Andor Technology, Belfast, UK).

Stress maps were calculated from the ERISM and WARP displacement maps as described previously (*Kronenberg et al., 2017b*) using an FEM simulation via COMSOL multiphysics (COMSOL Ltd, Cambridge, UK) and the known mechanical properties of the microcavity.

## Polydimethylsiloxane gel preparation

Polydimethylsiloxane elastomers were prepared according to the manufacturer's guidelines for all gels. The two component precursors of different gels were mixed together in separate glass bottles using an equal mass ratio of the two components for Sylgard 527 and NulSil Gel8100 but a 1:10 volumetric ratio for Sylgard 184. Mixing was performed by 10 min of magnetic stirring (Sylgard 527 and NuSil GEL8100) or by 10 min of mechanical stirring (Sylgard 184). The elastomer mixtures were then combined in a fresh bottle in the desired mass ratio using a syringe following the same method as a previous study (*Palchesko et al., 2012*). Combined elastomers were mixed for a further 10 min. Mixtures containing Sylgard184 were initially mixed by high-speed vortexing to coarsely disperse the gel to allow for the magnetic stir bar to overcome the high viscosity of the gel. After mixing, all preparations were degassed under vacuum for around 5 min, prior to fabrication of microcavities.

## Anaesthetised animal force imaging

Animals were selected, cleaned, and placed in a fridge at 4°C for 2–3 hr to anaesthetise them. Immediately prior to experiments, anaesthetised animals were gently placed onto a collagen-coated microcavity in a Petri dish on ice. The microcavities were then placed, using a moistened paint brush, on the ERISM-WARP microscope and the animals were observed carefully. As soon as mouthhook movement was observed, an ERISM measurement was taken. Animals often had to be placed back onto ice to anaesthetise them once more as they rapidly regained motility. As the complete ERISM scan requires ca. 5 s, animals were required to be completely stationary in order to obtain reliable stress map images.

## Freely behaving animals force imaging

Animals were selected according to the previously outlined criteria and cleaned before being placed onto a 1% (w/v) agarose-lined Petri dish. Elastic resonators were prepared according to the coating criteria mentioned above. 10% NusilGEL8100, 180 W $O_2$ plasma-treated microcavities were used for all freely behaving experiments. Once calibration images of the microcavity were acquired, excess water was removed from the cavity and animals were gently placed onto the cavity surface with a paintbrush, taking care to ensure there was enough moisture on the animal to prevent drying by wetting the paintbrush prior to transferring the animal. In order to keep animals on the sensor surface, a 50 µL drop of 15 mM ethyl butanoate (Sigma-Aldrich Inc, MO), suspended in paraffin oil, was dropped onto a 24 mm$^2$ glass coverslip before being inverted and placed on top of the silicone well (ibidi GmbH) such that the attractive odorant faced towards the animal but perpetually out of its reach. Animal substrate interaction was then imaged by WARP, using alternating wavelengths to generate a series of interleaved cavity resonance images, and displacement and stress maps were generated as described earlier. All WARP videos were recorded at 120 FPS, producing displacement maps with an effective frame rate of 60 FPS, using a ×4 magnification objective. Due to the high frame rate, we were limited to the use of ¼ of the total camera sensor, thus higher magnifications would prevent mapping of the whole animal.

## Statistical analyses

All statistical analyses were performed using OriginPro 2019 (OriginLab Corporation). Coefficients of determination ($R^2$) for all but GRF vs. contact area analysis were determined using a linear fit. The rarity of backward waves during normal larval behaviour precluded analysis of latencies as used in *Figure 2*. Adjusted coefficients of determination (Adj. $R^2$) for the GRF vs. contact area analysis were performed using a second-order polynomial fit instead as this describes the data better than a linear fit. Two-way repeated-measures ANOVA was used in segmentwise peak contact area

analysis as data were normally distributed according to a Shapiro–Wilk test. However, Levene's test for homogeneity of variances was significant for SI ($p<0.05$) but not for ST ($p=0.092$), we urge caution when interpreting the within-subjects' effects. Mauchly's test showed sphericity of segment ($W = 0.082$, $p=0.063$) and the segment * SI-ST interaction ($W = 0.27428$, $p=0.62463$), where the SI-ST factor was not tested due to insufficient degrees of freedom. Independent-samples $t$-test was performed to show no significant difference between larval behaviour on elastic resonators and standard agarose substrates as data were normally distributed according to a Shapiro–Wilk test. Pairwise comparisons between segments all used Tukey-corrected $t$-tests. Force–distance curves were fitted using a height-corrected Hertz model; all force–distance curves were fitted with an $R^2 > 0.9$.

## Acknowledgements

This work was supported by EPSRC (Doctoral Training grant EP/L505079/1 and grant EP/P030017/1), the European Research Council under the European Union's Horizon 2020 Framework Programme (FP/2014-202) ERC grant agreement no. 640012 (ABLASE), and the Alexander von Humboldt Foundation via the Humboldt Professorship to MCG. We thank our technicians Audrey Grant and Tanya Sneddon for preparing fly food, Dr Eleni Dalaka for her support during atomic force microscopy experiments, Dr Andreas Mischok for his advice with fabrication, Dr Marcus Bischoff for his advice and support early in the work, and Dr Jacob Francis and Dr James MacLeod for their methodological advice regarding lateral view imaging.

## Additional information

### Funding

| Funder | Grant reference number | Author |
|---|---|---|
| Engineering and Physical Sciences Research Council | EP/L505079/1 | Jonathan H Booth |
| Engineering and Physical Sciences Research Council | EP/P030017/1 | Malte C Gather |
| European Research Council | FP/2014-202 | Malte C Gather |
| European Research Council | 640012 (ABLASE) | Malte C Gather |
| Alexander von Humboldt Foundation | Humboldt Professorship | Malte C Gather |

The funders had no role in study design, data collection and interpretation, or the decision to submit the work for publication.

### Author contributions

Jonathan H Booth, Conceptualization, Resources, Data curation, Software, Formal analysis, Investigation, Visualization, Methodology, Writing – original draft, Writing – review and editing; Andrew T Meek, Conceptualization, Software, Methodology; Nils M Kronenberg, Conceptualization, Resources, Software, Validation, Methodology, Writing – review and editing; Stefan R Pulver, Conceptualization, Supervision, Validation, Investigation, Methodology, Writing – original draft, Project administration, Writing – review and editing; Malte C Gather, Conceptualization, Resources, Software, Formal analysis, Supervision, Funding acquisition, Validation, Investigation, Methodology, Writing – original draft, Project administration, Writing – review and editing

### Author ORCIDs

Jonathan H Booth http://orcid.org/0000-0002-9198-226X
Stefan R Pulver http://orcid.org/0000-0001-5170-7522
Malte C Gather http://orcid.org/0000-0002-4857-5562

Reviewer #1 (Public review): https://doi.org/10.7554/eLife.87746.3.sa1
Reviewer #2 (Public review): https://doi.org/10.7554/eLife.87746.3.sa2
Author response https://doi.org/10.7554/eLife.87746.3.sa3

## Additional files

### Supplementary files
• Supplementary file 1. Measured effective Young's modulus per elastomer mixture post plasma treatment.
• MDAR checklist

### Data availability
The research data underpinning this publication can be accessed at: https://doi.org/10.17630/f2a655d8-85c7-4bc0-b01b-b6f67aab2f07.

The following dataset was generated:

| Author(s) | Year | Dataset title | Dataset URL | Database and Identifier |
|---|---|---|---|---|
| Booth JH, Meek AT | 2024 | Optical mapping of ground reaction force dynamics in freely behaving *Drosophila melanogaster* larvae | https://doi.org/10.17630/f2a655d8-85c7-4bc0-b01b-b6f67aab2f07 | University of St Andrews Research Repository, 10.17630/f2a655d8-85c7-4bc0-b01b-b6f67aab2f07 |

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
