## [Editor Report · eLife assessment]

This study reports **important** findings about new locomotory dynamics of crawling *Drosophila* larva based on imaging the reaction forces during larval crawling. The evidence with the new high-resolution microscopy method is **compelling** as it significantly improves the spatial, temporal, and force resolution compared to previous methods for studying *Drosophila* larva and could be applied to other crawling organisms. The article explains the new technology, WARP microscopy, and provides analysis of the data to characterise small animal behaviour and discover new crawling-associated anatomical features and motor patterns. The work will be of interest to the broad neuroscience community interested in the mechanisms of locomotion in a highly tractable model.

---

## [Referee Report · Reviewer #1 (Public review)]

This work demonstrates a new technique to characterize the interaction between a crawling larva and the substrate on which it is crawling, at much higher temporal speed and spatial resolution than previously possible. While I have some questions about the interpretation of the data, both the demonstration of WARP microscopy to characterize small animal behavior and the discovery of new crawling-associated anatomical features and motor patterns make the paper worthy of attention.

I thank the authors for providing data underlying the figures. In these uncropped data sets, the deformation of the substrate due to the surface tension of an adhering water layer is visible. I would hope the authors would provide a subset of these images and some of the accompanying information (e.g. that the deformation of the gel due to the water layer cannot be accurately calculated due to too-rapid phase wrapping in the interferogram) as supplements to the text, to aid in interpretation and understanding of the data. It is also worth noting that in the data provided, under the larva, the integral of the stress on the gel is upward, despite the downward force exerted by the protopodia.

Future work using this exciting technique might address the role of surface tension and the balance of forces and might also produce direct evidence to show that the protopodia serve to "anchor" segments of the larva not in motion. Indeed, the most exciting aspect of this work is the number of new questions it both raises and provides a technological pathway towards resolving.

---

## [Referee Report · Reviewer #2 (Public review)]

The biology and dynamics is well-described. The ERISM and WARP methods are state-of-the-art. The most important new information is the highly accurate and detailed maps of displacement. The real achievements are the new locomotory dynamics uncovered with amazing displacement measurements. One key discovery is the broad but shallow anchoring of the posterior body when the anterior body undertakes a "head sweep". Another discovery is the tripod indentation at the tail at the beginning of peristalsis cycles. This paper describes the detailed dynamics of anchoring for the first time. Anchoring behavior now has to be included in the motor sequence for *Drosophila* larva locomotion in any comprehensive biomechanical or neural model.

---

## [Author Response]

The following is the authors’ response to the original reviews.

**Reviewer #1 (Public Review):**
(1.1) This work introduces a new method of imaging the reaction forces generated by small crawling organisms and applies this method to understanding locomotion of *Drosophila* larva, an important model organism. The force and displacement data generated by this method are a qualitative improvement on what was previously available for studying the larva, improving simultaneously the spatial, temporal, and force resolution, in many cases by an order of magnitude. The resulting images and movies are quite impressive.

We thank the reviewer for their recognition of the achievements our work presents and for their feedback with regard to what they consider our most important findings and the points raised in their review. We will address these points individually below.

(1.2) As it shows the novel application of recent technological innovations, the work would benefit from more detail in the explanation of the new technologies, of the rationales underlying the choice of technology and certain idiosyncratic experimental details, and of the limitations of the various techniques. In the methods, the authors need to be sure to provide sufficient detail that the work can be understood and replicated. The description of the results and the theory of motion developed here focus only on forces generated when the larva pushes against the substrate and ignores the equally strong adhesive forces pulling the larva onto the substrate.

As the reviewer correctly points out, our present work adapts a recently developed set of methods (namely, ERISM and WARP) for use with small soft-bodied animals. The foundational methods have been described in detail in previous publications (refs, 23 and 26). However, upon reflection, we agree that more information can be provided to ensure our work is more accessible and reproducible. We also agree that some additional clarifying information on our approach could be helpful. We have addressed this in the following ways:

(1) We have included a detailed Key Resources table in the methods section to allow for maximum transparency on equipment and reagent sourcing. This can now be found on Pages 16-19.

(2) We have modified the ‘Freely behaving animals force imaging’ section of the Materials and Methods section to include more detailed information on practical aspects of conducting experiments. These changes can be found on page 23-24 (lines 566–567, 571-577).

(3) We have re-ordered the Materials and Methods section, such that microcavity fabrication and microcavity characterisation occur prior to the description of ERISM and WARP experiments - this change should hopefully aid replication. Details regarding the application of a silicone well to the surface of microcavities have also been added (lines 472-474).

(4) We have added additional text in the Introduction and Results (Pages 3-4 and 7, lines 56-86, and 152-153) to explain our rationale for using ERISM/WARP and additional text in the discussion that discusses the potential role(s) of adhesive forces in larval locomotion (Page 12, lines 301307).

(1.3) The substrate applies upward, downward, and horizontal forces on the larva, but only upward and downward forces are measured, and only upward forces are considered in the discussions of "Ground Reactive Forces." An apparent weakness of the WARP technique for the study of locomotion is that it only measures forces perpendicular to the substrate surface ("vertical forces" in Meek et al.), while locomotion requires the generation of forces parallel to the substrate ("horizontal forces"). It should be clarified that only vertical forces are studied and that no direct information is provided about the forces that actually move the larva forward (or about the forces which impede this motion and are also generated by the substrate). Along with this clarification, it would be helpful to include a discussion of other techniques, especially micropillar arrays and traction force microscopy, that directly measure horizontal forces and of why these techniques are inappropriate for the motions studied here.

We attempted to provide a streamlined Introduction in our initial submission and then compared ERISM/WARP to other methods in our discussion. We are happy to provide a brief overview of substrate force measurement methods in the introduction to help set the stage for readers. The Introduction section of our revised manuscript now contains the following comparison of different mechanobiological imaging techniques on pages 3-4 lines 56-86:

‘However, in the field of cellular mechanobiology, many new force measuring techniques have been developed which allow measurement of comparatively small forces from soft structures exhibiting low inertia (15–17) often with relatively high spatial-resolution. Early methods such as atomic force microscopy required the use of laser-entrained silicon probes to make contact with a cell of interest (15). This approach is problematic for studying animal behaviour due to the risk of the laser and probe influencing behaviour. Subsequently, techniques have been developed which allow indirect measurement of substrate interactions. One such approach is Traction Force Microscopy (TFM) in which the displacement of fluorescent markers suspended in a material with known mechanical properties relative to a zero-force reference allows for indirect measurement of horizontally aligned traction forces (17–19). This technique allows for probe-free measurement of forces, but the need to obtain a precise zero-force reference would make time-lapse measurements on behaving animals challenging; further, depending on the version used, it has insufficient temporal resolution for the measurement of forces produced by many behaving animals, despite recent improvements (20). A second approach revolves around the use of micropillar arrays; in this technique, horizontally-aligned traction forces are measured by observing the deflection of pillars made of an elastic material with known mechanical properties. This approach can be limited in spatial resolution and introduces a non-physiological substrate that may influence animal behavior (21,22).

Recently we have introduced a technique named Elastic Resonator Interference Stress Microscopy (ERISM) which allows for the optical mapping of vertically aligned GRFs in the pico and nanonewton ranges with micrometre spatial resolution by monitoring local changes in optical resonances of soft and deformable microcavities. This technique allows reference-free mapping of substrate deformations and calculation of vertically directed GRFs; it has been used to study a range of questions related to exertion of cellular forces (23–25). Until recently, this technique was limited by its low temporal resolution (~10s), making it unsuitable for recording substrate interaction during fast animal movements, but a further development of ERISM known as wavelength alternating resonance pressure microscopy (WARP), has been demonstrated to achieve down to 10 ms temporal resolution (26). Given ERISM/WARP allows for probe-free measurement of vertical ground reaction forces with high spatial and temporal resolution, it becomes an attractive method for animal-scale mechanobiology.’

(1.4) The larvae studied are about 1 mm long and 0.1 mm in cross-section. Their volumes are therefore on order 0.01 microliter, their masses about 0.01 mg, and their weights in the range of 0.1 micronewton. This contrasts with the force reported for a single protpodium of 1 - 7 micronewtons. This is not to say that the force measurements are incorrect. Larvae crawl easily on an inverted surface, showing gravitational forces are smaller than other forces binding the larva to the substrate. The forces measured in this work are also of the same magnitude as the horizontal forces reported by Khare et al. (ref 32) using micropillar arrays.I suspect that the forces adhering the larva to the substrate are due to the surface tension of a water layer. This would be consistent with the ring of upward stress around the perimeter of the larva visible in S4D, E and in video SV3. The authors remark that upward deflection of the substrate may be due to the Poisson's ratio of the elastomer, but the calibration figure S5 shows that these upward deflections and forces are much smaller than the applied downward force. In any case, there must be a downward force on the larva to balance the measured upward forces and this force must be due to interaction with the substrate. It should be verified that the sum of downward minus upward forces on the gel equals the larva's weight (given the weight is neglible compared to the forces involved, this implies that the upward and downward forces should sum to 0).

We have carefully calculated the forces exerted by protopodia and are confident in the accuracy of our measurements as reported. We further agree with the reviewer’s suggestion that gravitational forces can be largely neglected.

As the reviewer points out, one would expect forces due to upward and downward deflections to cancel when considering the entire system. However, we see indications that the counteracting / balancing force often acts over a much larger area than the acting force, e.g. a sharp indentation by a protopodium might be counteracted by an upward deflection over a 10-20 fold larger radius and hence 100 to 400-fold larger area, thereby reducing the absolute value of the upward deflection at any given pixel surrounding the indentation. This in turn increases error in determining the integrated upward deformation, making it difficult to perform an absolute comparison of acting and counteracting force. Further, recording the entire counteracting force induced deformation would require acquiring data with a prohibitively large field of view.

We agree that in some situations, water surface tension may be adhering animals to the substrate. Importantly, this is a challenge that the animal faces outside the lab in its natural environment of moist rotting fruit and yeast. The intricate force patterns seen in our study in the presence of water surface tension are therefore ecologically relevant. In other situations (e.g. preparing for pupation), larvae are able to stick to dry surfaces, suggesting that other adhesive forces such as mucoid adhesion can also come into play in certain behavioural contexts. A full characterization of the effects of water tension and mucoid adhesion are beyond the scope of this study. However, we have now added a sentence on pages 8 and 12 commenting on these other biomechanical forces at play:

‘We also observed that the animals travel surrounded by a relatively large water droplet (lines 189-190).’

‘We observed that larvae travel surrounded by moisture from a water droplet, which produces a relatively large upwardly directed force in a ring around the animal. The surface tension produced by such a water droplet likely serves a role in adhering the animal to the substrate. However, during forward waves, we found that protopodia detached completely during SwP, suggesting this surface tensionrelated adhesion force can be easily overcome by the behaving animal. (lines 301-307) .’

(1.5) Much of the discussion and the model imply that the sites where the larva exerts downward force on the gel are the sites where horizontal propulsion is generated. This assumption should be justified. Can the authors rule out that the larva 'pulls' itself forward using surface tension instead of 'pushing' itself forward using protopodia?

Determining the exact ‘sites’ where horizontal propulsion is generated is challenging. In our conceptual model, movement is not initiated by protopodia per se, but rather by a constellation of muscle contractions, which act upon the hydrostatic skeleton, which in turn causes visceral pistoning that heaves larvae forward. This is based on previous findings in Ref 31. While there are indeed downward protopodial ‘vaulting’ forces prior to initiation of swing, we propose that the main function of protopodia is not to push the larvae forward, but rather to provide anchoring to counteract opposing forces generated by muscles. We agree that water surface tension could also be sculpting biomechanical interactions; however, a full characterization of how water surface tension shapes larval locomotion is beyond the scope of this study.

Since we have observed larvae move over dry terrain (e.g. glass) without an encasing water bubble, we do not believe that an encasing water bubble is strictly required for locomotion. We have also seen no obvious locomotion related modulations in the pulling forces created by water bubbles encasing larva, which would be expected if animals were somehow using water tension to pull themselves forward. Overall, the most likely explanation is that larvae use a mixture of biomechanical tactics to suit the moment in a given environment. This represents a challenge but also an opportunity for future research.

We have now added additional text in the ‘Functional subdivisions within protopodia’ subsection to discuss these nuances (page 14, lines 382-387):

‘This increased force transmitted into the substrate is unexpected as the forces generated for the initiation of movement should arise from the contraction of the somatic muscles. We propose that the contraction of the musculature responsible for sequestration acts to move haemolymph into the protopodia thus exerting an increased pressure onto the substrate while the contact area decreases as a consequence of the initiation of sequestration.’

and (page 15, lines 398-399):

‘Water surface films appear to facilitate larval locomotion in general but the biomechanical mechanisms by which they do this remain unclear.’

(1.6) More detail should be provided about the methods, their limitations, and the rationale behind certain experimental choices.

We thank the reviewer for this comment. As this significantly overlaps with a point raised earlier, we kindly direct them to our answer to comment #1.2 above.

(1.7) Three techniques are introduced here to study how a crawling larva interacts with the substrate: standard brightfield microscopy of a larva crawling in an agarose capillary, ERISM imaging of an immobilized larva, and WARP imaging of a crawling larva. The authors should make clear why each technique was chosen for a particular study - e.g. could the measurements using brightfield microscopy also be accomplished using WARP? They should also clarify how these techniques relate to and possibly improve on existing techniques for measuring forces organisms exert on a substrate, particularly micropillar arrays and Traction Force Microscopy.

Indeed, each of the three methods used has a specific merit. The brightfield microscopy was selected to track features on the animal’s body and to provide a basic control for the later measurements. However, this technique cannot directly measure the substrate interaction, it only allows inferences to be made from tracked features at the substrate interface. ERISM provides high resolution maps of the indentation induced by the larva; it is also extensively validated for mapping cell forces and the data analysis is robust against defects on the substrate (refs 23, 24 and 25). However, as we explain in the manuscript, ERISM lacks the temporal resolution needed to monitor mechanical activity of behaving larva. Its use was therefore limited to the study of anaesthetised animals. For mapping forces exerted by behaving larva, we used WARP which is a further development of ERISM that offers higher frame rates but at the cost of requiring more extensive calibration (Supplementary Figure S4). The streamlined introduction of the different methods in our original manuscript originates from our attempt to be as concise as possible. However, as state in response to comment #1.2, we agree that additional explanation and discussion will be helpful for readers and that it will helpful to briefly refer to other methods for force mapping. We have now added references to a variety of techniques in the Introduction (Page 3-4, lines 56-86) as stated in a prior response.

(1.8) As written, "(ERISM) (19) and a variant, Wavelength Alternating Resonance Pressure microscopy (WARP) (20) enable optical mapping of GRFs in the nanonewton range with micrometre and millisecond precision..." (lines 53-55) may generate confusion. ERISM as described in this work has a much lower temporal resolution (requires the animal to be still for 5 seconds - lines 474-5); In this work, WARP does not appear to have nanonewton precision (judging by noise on calibration figures) and it is not clear that it has millisecond precision (the camera used and its frame rate should be specified in the methods).

Previous studies have demonstrated the capabilities and limitations of ERISM and WARP. Upon reflection, we agree that our wording here could be more precise. To clarify our claim, we now separate the statements on ERISM and WARP in the introduction as follows (page 4, lines 78-83):

“Until recently, this technique was limited by its low temporal resolution (~10s) making it unsuitable for use in recording substrate interaction during fast animal movements, but a further development of ERISM known as wavelength alternating resonance pressure microscopy (WARP), has been demonstrated to achieve down to 10 ms temporal resolution (26)”

While WARP can achieve comparable force resolution as ERISM when used in a cellular context (c.f. Ref 26), we agree that for the present study, the resolution was in the 10s of nanonewton range, due to the need to use stiffer substrates and larger fields of view.

The camera used in our work was specified in the appropriate subsection of the Materials and Methods (“All WARP and ERISM images were acquired using an Andor Zyla 4.2 sCMOS camera (Andor Technology, Belfast, UK)”). We apologise that the exact frame rate used in our current work was not mentioned in our original manuscript; this has now been added to the ‘Freely behaving animals force imaging’ section of the Materials and Methods (page 23, lines 574-577).

(1.9) It would be helpful to have a discussion of the limits of the techniques presented and tradeoffs that might be involved in overcoming them. For instance, what is the field of view of the WARP microscope, and could it be increased by choosing a lower power objective? What would be required to allow WARP microscopy to measure horizontal forces? Can a crawling larva be imaged over many strides by recentering it in the field of view, or are there only particular regions of the elastomer where a measurement may be made?

We agree with the reviewer that some discussion of the limitations of our technique will allow readers to have a more informed appreciation of what we are capable of measuring using WARP. However, as this is the first work to ever demonstrate such measurements, the limitations and tradeoffs cannot all be known with certainty at the present stage.

To answer your individual questions:

(1) There is a trade-off between numerical aperture and the ability to resolve individual interference fringes. Since our approach to calculate displacement from reflection maps relies upon counting of individual fringe transitions, going to a lower powered objective risks having these fringes blend and thus the identification of the individual transitions becoming impossible. The minimum numerical aperture of the objective will therefore generally depend on the steepness of indentations produced by the animals; the steeper an indentation, the closer the neighbouring fringes and thus the higher the required magnification to resolve them.

(2) From WARP and ERISM data, one can make inferences about horizontal forces, as is described in detail in our earlier publications about ERISM (ref, 23). However, quantitation of horizontal forces at sufficient temporal resolution to allow the investigation of behaving *Drosophila* larva is currently not possible.

(3) Many strides can indeed be imaged using our technique, however, this comes with additional technical challenges. Whether or not the animal itself can be recentred is an ongoing challenge. We have found that the animals are amenable to recentring themselves within the field of view if chasing an attractive odorant. However, manual recentering using a paintbrush risks destroying the top surface of the soft elastic resonator and recentering the microscope stage would require real-time object tracking which has been outside the scope of this original work, given the other challenging requirements on hardware and optics for obtaining high quality force maps.

To provide more information on limitations of our technique, we have added the following text into the discussion (pages 13-14, lines 356-370).

‘Despite the substantial advances they have provided, the use of WARP and ERISM also brings challenges and has several technical limitations. For example, fabrication of resonators is much more challenging than preparation of the agarose substrates conventionally used for studying locomotion of *Drosophila*. This problem is compounded by the fragility of the devices owing to the fragility of the thin gold top mirror. This becomes problematic when placing animals onto the microcavities, as often the area local to the initial placement of the animal is damaged by the paintbrush used to move the animals. Further, as a result of the combining of the two wavelengths, the effective framerate of the resultant displacement and stress maps is equal to half of the recorded framerate of the interference maps. To be able to monitor fast movements, recording at very high framerates is therefore necessary which, depending on hardware, might require imaging at reduced image size, but this in turn reduces the number of peristaltic waves that can be recorded before the animal escapes the field of view. A further limitation is that WARP and ERISM are sensitive mainly to forces in the vertical direction; this is complementary to TFM, which is sensitive to forces in horizontal directions. Using WARP in conjunction with high speed TFM (possibly using the tuneable elastomers presented here) could provide a fully integrated picture of underlying vertical and horizontal traction forces during larval locomotion.’ And further on page 13, lines 337-341:

‘More detailed characterisation of this behaviour remains a challenge owing to the changing position of the mouth hooks. Due to their rigid structure and the relatively large forces produced in planting, mouth hooks produce substrate interaction patterns which our technique struggles to map accurately due to overlapping interference fringes ambiguating the fringe transitions.’

We trust that the above discussion and our modifications to our manuscript resulting from these will address the reviewer’s concerns.

**Reviewer #2 (Public Review):**
(2.1) With a much higher spatiotemporal resolution of ground dynamics than any previous study, the authors uncover new "rules" of locomotory motor sequences during peristalsis and turning behaviors. These new motor sequences will interest the broad neuroscience community that is interested in the mechanisms of locomotion in this highly tractable model. The authors uncover new and intricate patterns of denticle movements and planting that seem to solve the problem of net motion under conditions of force-balance. Simply put, the denticulated "feet" or tail of the *Drosophila* larva are able to form transient and dynamic anchors that allow other movements to occur.

We thank the reviewer for their feedback and the information regarding which of our results is likely to resonate most impactfully with readers from a biological background.

The biology and dynamics are well-described. The physics is elementary and becomes distracting when occasionally overblown. For example, one doesn't need to invoke Newton's third law, per se, to understand why anchors are needed so that peristalsis can generate forward displacements. This is intuitively obvious.

We are sorry to hear that the reviewer found some of the physics details distracting. To address this concern, we have simplified some of the language while still attempting to keep the core arguments intact. For context and analogy, we still believe that including a brief reference to the laws of motion is helpful for some readers to explain some of our results and highlight their general implications, especially with regard to anchoring against reaction forces.

One of our objectives is to make this article accessible and interesting for biologists and physicists at all levels. We feel it is important to reach out to both communities and try to be inclusive as possible in our writing. Newton’s 3rd law is clearly relevant for our study and it is a common point of reference for anyone with a highschool education, and so we feel it is appropriate to mention it as a way to help readers across disciplines understand the biophysical challenges faced by the animals we study.

(2.2) Another distracting allusion to "physics" is correlating deformation areas with displaced volume, finding that "volume is a consequence of mass in a 2nd order polynomial relationship". I have no idea what this "physics" means or what relevance this relationship has to the biology of locomotion.

Upon reflection, we agree that this language may be overly complex and distracts from what is, at its core, a simple, but important principle governing how *Drosophila* larvae interact with their substrates. The point we are trying to make is that our data show that forces exerted by an animal are proportional in a non-linear way to contact area. This suggests that to increase force exerted on the substrate, an animal must increase contact area. We do not observe contact area remaining constant while force increases, or vice versa. To make this result more clear, we have made several changes in our revised manuscript. Figure 5B no longer shows the relationship between the protopodial contact area and the displaced volume of the elastic resonator, but instead now shows the protopodial contact area and recorded force transmitted into the substrate. This then shows that in order to increase force transmitted into the substrate, these animals must increase their contact area. We have made changes to the figure legend of Figure 5 and the statements in the Results section accordingly (Page 9, lines 220-222).

2.3 The ERISM and WARP methods are state-of-the-art, but aside from generally estimating force magnitudes, the detailed force maps are not used. The most important new information is the highly accurate and detailed maps of displacement itself, not their estimates of applied force using finite element calculations. In fact, comparing displacements to stress maps, they are pretty similar (e.g., Fig 4), suggesting that all experiments are performed in a largely linear regime. It should also be noted that the stress maps are assumed to be normal stresses (perpendicular to the plane), not the horizontal stresses that are the ones that actually balance forces in the plane of animal locomotion.

We largely agree with the statement made by the reviewer here. However, we have found that in many contexts, audiences appreciate having the absolute number of the forces and stresses involved reported. Therefore, where possible, we have used stress maps, rather than displacement maps. We also observe that while stress and displacement maps show similar patterns, features sometimes appear sharper in the stress map, which is a result of the finite element algorithm being able to attribute a broad indentation to a somewhat more localised downward force. We have thus opted to keep to original stress maps. We have been more explicit about WARP and ERISM being more tuned to recording vertically directed forces throughout the revised manuscript (lines 75, 78, 86, 162, 301, 305, 336).

We have also modified our Discussion section to encourage further investigation of our proposed model using a technique more tuned to horizontal stresses (pages 12-13, lines 324-328):

‘However, WARP microscopy is best suited to measurements of forces in the vertical direction, and though we can make inferences such as this as they are a consequence of fundamental laws of physics, we present this conclusion as a testable prediction which could be confirmed using a force measurement technique more tuned to horizontally directed forces relative to the substrate.’

(2.4) But none of this matters. The real achievements are the new locomotory dynamics uncovered with these amazing displacement measurements. I'm only asking the authors to be precise and down-to-earth about the nature of their measurements.

We thank the reviewer for their perceptiveness in finding that though the forces are interesting, the interactions themselves are the most noteworthy result here. We trust that with the changes made in our revised manuscript, the description is now more “down-to-earth”, more concise where appropriate, and accurate as to which results are particularly important and novel.

(2.5) It would be good to highlight the strength of the paper -- the discovery of new locomotion dynamics with high-resolution microscopy -- by describing it in simple qualitative language. One key discovery is the broad but shallow anchoring of the posterior body when the anterior body undertakes a "head sweep". Another discovery is the tripod indentation at the tail at the beginning of peristalsis cycles.

We thank the reviewer for this recommendation. We agree that including a more explicit statement of some of our findings, especially with regards to these new posterior tripod structures and the whole-abdomen preparatory anchoring prior to head sweeps, would make the paper more impactful. As a result, we have modified the discussion section to include a statement for each new result and have also amended our abstract as a result (lines 407-416):

“Here we have provided new insights into the behaviour of *Drosophila* larval locomotion. We have provided new quantitative details regarding the GRFs produced by locomoting larvae with high spatiotemporal resolution. This mapping allowed the first detailed observations of how these animals mitigate friction at the substrate interface and thus provide new rules by which locomotion is achieved. Further, we have ascribed new locomotor function to appendages not previously implicated in locomotion in the form of tripod papillae, providing a new working hypothesis of how these animals initiate movement. These new principles underlying the locomotion outlined here may serve as useful biomechanical constraints as called for by the wider modelling community (39).”

(2.6) As far as I know, these anchoring behaviors are new. It is intuitively obvious that anchoring has to occur, but this paper describes the detailed dynamics of anchoring for the first time. Anchoring behavior now has to be included in the motor sequence for *Drosophila* larva locomotion in any comprehensive biomechanical or neural model.

We agree with the reviewer on this. We think it is best to let our colleagues reflect on our findings and then decide how best to include them in future models.

**Recommendations for the authors:**

**Reviewer #1 (Recommendations For The Authors):**
Please be sure to describe in a figure caption or in the methods the details of the optical setup, especially the focal lengths of all the lenses, including the objective, and part numbers of the LEDs and filters. It would be helpful to have a figure in the main paper explaining the principles of ERISM/WARP microscopy along with the calibration measurements and computational pipeline (this would mainly combine elements already in the supplement). Such a figure should also include details of the setup that are alluded to in the methods but not fully explained (for instance, a "silicone well" is referred to in the methods but never described). The calibration of elastomer stiffness that now appears in the main text could be made a supplementary figure, unless there is some new art in the fabrication of the elastomers that should be highlighted as an advance in the main text.

We appreciate the importance of explaining our methods to readers.

In response to the public comments, we have added further details in our methods section to clarify practical aspects and ensure that readers will be able to reproduce our work.

In Supplemental Figure 2, we show the full optical light path for ERISM and WARP along with named components. In addition, the principles of ERISM and WARP microscopy have already been extensively described in previous publications (See Refs 23-26). In light of this, we feel that the best approach in this paper is to direct readers to those publications.

We feel that it is appropriate to present the calibration of elastomer stiffness in the main text because this is indeed a new innovation that is not just about making the elastomers but making force sensors based on these different materials. This is really important because it shows how researchers can tune the stiffness of an ERISM/WARP elastomer to match the type of tissue or organism under study. This is really the key technical advance that enables whole animal biomechanics across a range of animal sizes, so we think it is appropriate to keep it in the main text.

We want to make sure that we do not oversell this point, and we feel that we make it sufficiently clear in the main text of our manuscript that making elastomer based force sensors of appropriate stiffness is important, when we state

“First, we developed optical microcavities with mechanical stiffnesses in the range found in hydrogel substrates commonly used for studying *Drosophila* larval behaviour, i.e. Young’s modulus (E) of 10-30kPa (36–38).” (p. 5, ll. 124) and later

“Here we used *Drosophila* larvae as a test case, but our methods now allow elastic optical resonators to be tuned to a wide range of animal sizes and thus create new possibilities for studying principles of neuro-biomechanics across an array of animals.” (p. 12, ll. 337)

I would appreciate a description of the "why" behind some experimental choices, as understanding the motivation would be helpful for other researchers looking to adopt these techniques.

We have now added additional text in the introduction and discussion that explains the rationale behind our experimental choices. in more detail. Please see our response to Reviewer 1’s public comments on the same point.

(1) The WARP and ERISM experiments were conducted on a collagen coated gold surface rather than agarose. Why? EG does agarose not adhere to the gold, or would its thickness interfere with the measurement?

The gold layer is applied above the elastomer and the collagen on top of the gold layer makes the gold a more natural biological surface for the animals. Agarose is unsuitable as an elastomer because it would dry during the vacuum based deposition of the gold. It is also unsuitable as a surface coating on top of the gold as the coating on the gold needs to very thin to preserve the spatial and mechanical resolution of our sensors. Further, processing of agarose generally requires temperatures of 60°C and higher which we find can damage the elastomer / gold films.

(2) The ERISM measurements are made on a cold anesthetized animal right as it starts to wake up (visible mouth-hooks movement), which presents some difficulty. Why not start imaging while the animal is still completely immobile? Or why not use a dead larva?

This approach allowed us to get measurements of forces exerted by denticles that are physiologically and biomechanically accurate. In dead or fully anesthetized animals, one cannot be sure that the forces exerted by denticles and denticle bands are representative of the forces exerted by an animal with active hydrostatic control.

(3) In the ERISM setup the monochromator is spatially filtered by focusing through pinhole, while in the WARP setup, the LEDs are not.

Yes that’s correct. The LED light sources used in WARP have better spatial homogeneity than the tungsten filament used in ERISM and so a pinhole is not required in WARP.

(4) SV4 shows the interference image of a turning larva (presumably from one illumination wavelength) rather than a reconstruction of the displacement or stresses. Why?

We felt that in this particular case the interference images provided a clearer representation of the behavioural sequence, showing both the small indentations generated by individual denticles and the larger indentations of the animal overall.

Lines 49-50 "a lack of methods with sufficient spatiotemporal resolution for measuring GRFs in freely behaving animals has limited progress." This needs a discussion of what sufficient spatial and temporal resolutions would be and how existing methods fall short of these goals.

We have now rewritten the introduction to include an overview of other alternative approaches and of what we see as the requirements here. See our response to the public comments.

Figure caption 1B (line 789) refers to "concave areas of naked cuticle (black line) which generally do not interact with the substrate" While I think this might be supported by later WARP images, it's not clear how the technique of figure 1 measures interaction, which could e.g. be mediated by surface tension of a transparent fluid.

The technique of Figure 1 provides qualitative information which as the reviewer points out is validated by WARP measurements later.

Lines 184-189 "However, unexpectedly, we observed an additional force on the substrate when protopodia leave the substrate (SI) and when they are replanted (ST). To investigate whether this force was due to an active behaviour or due to shifting body mass, we plotted integrated displacement (i.e. displaced volume) against the contact area for each protopodium, combining data from multiple forwards waves (Figure 5B). Area is correlated with displaced volume for most time points, indicating that volume is a consequence of mass in a 2nd order polynomial relationship." I couldn't follow this argument at all.

We have now reworded this section and explained our rationale. Also see our response to a similar critique in Reviewer 2’s public comments.

Generally the authors might reconsider their use of acronyms. e.g. (244-246) "SI latencies were much more strongly correlated with wave duration across most segments than ST latencies. SIs scale with SwP and this could be mediated by proprioceptor activity in the periphery" is made more difficult to parse by the abbreviations.

As we need to refer to these terms multiple times throughout the manuscript, we feel the use of acronyms is appropriate here.

The video captions are inadequate. Please expand on them to explain clearly what is shown, and also describe in the methods how the data were acquired and processed. For instance, it seems that in SV3 a motion correction algorithm is applied so that the larva appears stationary even as it crawls forward. I think "fourier filtered" means that the images were processed with a spatial high pass filter - this should be explained and the parameters noted.

We have revisited the video captions provided in the supplementary information document and conclude that these contain the important information. The mode of acquisition are described in the methods, e.g. Video 1 and 2 see section in Methods on “Denticle band kinematic imaging” and Videos 3 and 4 see section in Methods on WARP. Supplementary Video 3 does not make use of motion correction; indeed, one can see the larvae moving upwards/forwards in the field of view. We apologize for not explaining the Fourier filtering process for Video 3. We have now modified the video caption to read as follows:

Video SV3. WARP imaging during forwards peristalses.

Video showing high frame rate displacement maps produced by a freely behaving *Drosophila* larva. Displacement maps were Fourier filtered to make denticulated cuticle more readily visible and projected in 3D to show the effects of substrate interaction. Details of the Fourier filtering procedure were described elsewhere [Kronenberg et al, Nat Cell Biol 19, 864–872 (2017)].

What were the reflectances of the bottom (10 nm Au/Cr) and top (15nm Au) metal layers at the wavelengths used? I imagine the bottom layer should be less than 38%, the top layer higher, and the product of the square of the bottom transmission and the top reflectance coefficients equal to the bottom reflectance (to make the two paths of the interferometer contribute equal intensity), but none of this is stated.

The reflectance of the gold mirrors was studied in detail in prior work on ERISM. See Kronenberg et al, Nat Cell Biol 19, 864–872 (2017). We therefore refrained from adding a complete optical characterization of the ERISM sensors again here. In brief, we found that a reflectance >13% at each Au mirror is required for reliable ERISM measurements.

The description of the gold coated elastomer as a microcavity is confusing to me. Does the light really make multiple round trips between the plates before returning to the detector? The loss of light on each round trip would depend on the reflectance and parallelism of the top and bottom mirrors. From the WARP calculation it's appears that there is only one round trip - a pi/2 phase shift results from the calculation for one round trip: 2pi*2nL 5nm/(630nm)^2, with n = 1.4 and L = 8 microns - if there were two round trips, the phase shift would be pi etc. Would this better be described as a mostly common path interferometer?

The physics of our devices is best described within the framework of thin film interference and (weak) microcavity optics. Indeed, light can make multiple roundtrips, though it gets attenuated with each reflection. The complete calculation of the multiple roundtrips is only required to obtain quantitative information on the amount of light that is reflected. The spectral position of minima in reflectance can also be obtained from assuming one roundtrip which is what is done in the description of the WARP calculations.

Figure 2 e,f: the line fits appear to be dominated by the data points at 2 s. If these are removed, do the fits change? To support the argument that 2e shows a correlation and 2f does not, some kind of statistical test, ideally a hierarchical bootstrap, should be conducted to compare between the two measurements.

If we remove the data points at 2 s, then R^2’s for swing initiation latencies change as follows: A2: 0.35 to 0.005; A4: 0.78 to 0.31; A6: 0.61 to 0.01. The data in 2e,f are the averages from 3 waves in each animal and so the data points at 2 s are not simply the result of single ‘rogue’ waves but rather averages of several trials. Further, if all individual waves are plotted, we can see that the overall trends are still visible.

We don’t think it is appropriate to remove the data at 2 s from our analysis, but we take the point regarding statements about presence or absence of correlation in a formal sense. We have therefore changed the wording in the description of 2e,f to refer simply to the fact that wave duration can ‘largely determine' latencies in some instances, but is less able to in other instances, as is suggested by the R^2 (coefficient of determination) data. In discussion, we have also adjusted our wording.

Figure 4 - please provide in the main figure or as a supplement the full images (i.e. not cropped to the assumed shape of the larva)

We do not feel that it is necessary or helpful to provide the full images given that the focus of the analysis is on dynamics of protopodia movements.

Figure 5e top: single data points around wave duration 0.6s appear to dominate fit lines. Does removing these points alter the fits? To support the argument that 5e top shows a correlation and 5e bottom does not, some kind of statistical test, ideally a hierarchical bootstrap, should be conducted to compare between the two measurements.

In Figure 5e, we are showing all waves analysed across animals. If we remove thedatapoints at 0.6 s, A2 R^2 changes from 0.24 to 0.05, A4 R^2 changes from 0.48 to 0.11, A6 R^2 changes from 0.69 to 0.34; however we don’t feel it is appropriate to remove these data from our analysis. We take the point about needing to be cautious about making claims about correlation versus no correlation and have now reworded description of these results along same lines as Figure 4.

It appears from the methods (467-489) that animals were kept wet for warp imaging but not for ERISM imaging. Please confirm or explain further the presence or absence of a water layer in these two sets of measurements, as this could affect the adhesion forces.

In each case, the animals were transferred onto experimental substrates with a moistened paintbrush. We have added text explicitly stating this in the methods section.

Kim et al. Nature Methods 2017 (10.1038/nmeth.4429) describes recording two images separated by less than 60 microseconds using a scientific CMOS camera with a frame rate of 200 Hz. This is accomplished by triggering a pulsed LED once at the end of one frame's capture window and then a second time at the beginning of the next frame's window (see Supplementary Figure 10). I'm not sure if this trick is widely known, but it's worth considering if the authors are running into a problem with movement between the two wavelength exposures in their WARP setup.

Thank you for this tip. We will take this under consideration for future work.

Is the setup compatible with optogenetics? (EG is the red light dim enough that it wouldn't activate CsChrimson, or could a longer wavelength led be used for interferometry?) If so, activation of mooncrawler descending neuron (MDN) could be used to study backward crawling (or thermogenetic activation of MDN), e.g. to contrast the sites and order of "anchoring" between the two directions of crawling.

The set-up is potentially compatible with optogenetics. We are in the process of exploring this in current ongoing work.

**Reviewer #2 (Recommendations For The Authors):**
Simplify/reduce the commentary about force measurements, and highlight the clear, qualitative descriptions of the novel locomotion patterns that they have observed. The microscopy and movements seem to matter more than the ground force estimations.

We have addressed these issues in our responses to Reviewer 2’s public comments.